# Cutting Process Consideration in Dynamic Models of Machine Tool Spindle Units

**Yurii Danylchenko [1], Michael Storchak [2],*, Mariia Danylchenko [1] and Andrii Petryshyn [1]**

[1]  Igor Sikorsky Kyiv Polytechnic Institute, Beresteisky Ave., 37, 03056 Kyiv, Ukraine;
    danilchenko.yu@gmail.com (Y.D.)

[2]  Institute for Machine Tools, University of Stuttgart, Holzgartenstraße 17, 70174 Stuttgart, Germany

*   Correspondence: michael.storchak@ifw.uni-stuttgart.de; Tel.: +49-711-685-83831

**Abstract:** Reducing the deviation effect from the specified machining conditions on the quality of the process in real time is the desired result of the intelligent spindle control system. To implement such a control system, a dynamic interaction model of the technological machining system with the cutting process was developed. The transfer matrix method of a multibody system was used in the development of the dynamic model. The physical closure condition of the technological machining system for using the transient matrix method is implemented in the developed model by introducing into this model an additional elastic coupling of the contact between the tool and the machined workpiece. The model is presented as a dynamic model of the elastic system "spindle unit–workpiece/tool–cutting process–tool/workpiece". To develop the dynamic model, the system decomposition was performed with an analytical description of the joint deformation conditions of the subsystems and the use of the transient matrix method to calculate the harmonic influence coefficients of these subsystems. The proposed approach is used to calculate the native vibration frequencies of the spindle with the workpiece fixed in the chuck and supported with the tool. The calculation results correspond to the experimental ones and quite accurately represent their trends for different contact interaction conditions.

**Keywords:** machine tool; spindle unit; cutting; dynamic model; transfer matrix method; multibodysystem





## 1. Introduction

The development of automated production is based on the operation of cyber–physical systems, which allow to observe and control the physical production process based on the results of virtual simulation using digital twins [1]. Digital twins combine real physical systems with an appropriate virtual representation and serve primarily to model and optimize production processes [2]. At the same time, the main effort in the development of digital twins of machining processes is aimed at reducing the gap between virtual simulation and real physical processes [3,4]. Machining equipment with the ability to monitor and control multiple process modules is called "smart machine tools" or "intelligent machine tools" [5]. The spindle units of these machines are called "intelligent spindles" [6]. Intelligent spindle development is based on research in six main areas, including tool state monitoring and control, vibration, spindle damage, temperature/thermal error, spindle balancing, and spindle durability. The basis of the smart spindle control strategy is to reduce or exclude the impact deviations from the set machining conditions on real-time machining quality. In this case, processing error compensation methods are based on measurements and models.

A critical review of the historical development, main problems, and trends in the design and functioning of the spindle unit as an integral quality assurance system is presented in a study by Abele et al. [7]. The modeling and analysis goal of spindle units are defined according to the authors as a simulation of spindle operation and optimization of

its design parameters. This is performed at the design stage to maximize dynamic stiffness and increase the material removal capacity with minimal size and minimum power consumption. This simulation is supposed to determine the dynamic characteristics of the mechanical system of the spindle unit. The spindle design and the factors associated with its configuration, installation, and operating conditions must be taken into account. Altintas and Cao considered spindle assemblies as an integral element of the mechanical part of the machine tool dynamic system [8]. Therefore, their dynamic characteristics depend on the character of mechanical connections with other system elements. Bach [9] and Zhang with colleagues [10] considered spindle assemblies carrying a workpiece or tool as complex mechanical oscillating systems with periodically changing mass–inertial characteristics and, accordingly, changing dynamic characteristics. In this case, the effect of different workpieces and mandrels with different tools were investigated. Hentati et al. [11] and Hu et al. [12] found a direct dependence on the machining accuracy of the spindle bearing system, which directly affects the dynamic characteristics of the machining system. Further studies in this field have established the existence of several zones of local resonances in the operating frequency range of spindle units [13,14]. The authors associate this with the proximity of the natural and forced vibrations caused by the cutting process dynamics. In addition, it has been found that this phenomenon is also caused by bearing defects [15]. When selecting the spindle unit design scheme and optimizing the spindle constructive parameters at the design stage, the spindle unit is usually represented as a "spindle–bearings" system, and its dynamic model is represented as a set of spindle dynamic models and bearings [16,17]. The finite element method (FEM) is most commonly used to build a dynamic spindle model. In this case, the spindle is represented as an elastic axisymmetric shaft with a distributed mass, modeled by Euler-Bernoulli, Rayleigh, or Timoshenko beam elements [16,18]. Using FEM allows to obtain the most accurate simulation results, but the creation of the model is associated with significant time consumption and complexity.

Among other ways to describe the spindle unit dynamics is to use the transfer matrix method [19]. The general approach to the application of TMM for determining the modes of bending and torsion of beams is presented in [20]. This method is based on solving the problem of bending eigenmodes of a beam element with a constant cross-section and a uniformly distributed mass. Compared to FEM, TMM has a few advantages. In particular, the direct characteristics of the model are involved in the TMM calculations [21]. These include stiffness and damping coefficients. These coefficients can be determined experimentally or from reference data. In addition, using the TMM method does not require solving a large equation system with a substantial number of beam model elements. The main advantage of using TMM is the simplicity of creating a computational dynamic model for the spindle unit with a sufficiently high calculation accuracy. However, the known models of spindle units [21–23] built using TMM are structurally limited to describing only the "spindle–bearings" system.

The representation of the spindle unit as a constituent element of the mechanical part from the machine tool dynamic system can be solved using the TMM method for a multibody system [24]. The main idea of the transfer matrix method of multibody systems (MSTMM) is the initial decomposition of a complex system into elements with simple dynamic characteristics [25]. For systems consisting of elastic and rigid components, the MSTMM reduces the dynamics problem to a general transfer equation [26,27]. This equation includes only the boundary state vectors. There is no need to repeatedly output most of the transfer matrices with simple elements, since the most common matrices are provided in the transfer matrix library [28].

A further development of the MSTMM method for controlled multibody systems is the discrete time transfer matrix method for controlled multibody systems (CMS) [29]. The method's peculiarity is the boundary state vectors forming by taking into account the control and back-coupling parameters on the results of actuator motion parameters tracking.

The main task of controlling machining processes on metal cutting machines is to ensure the stability of the cutting process by limiting the maximum cutting depth. Boundary cutting modes are determined by modeling the dynamic interaction between the cutting process and the machine structure [14,30,31]. The cutting process is mainly described by single-mass models with one (SDOF) or two (2SDOF) degrees of freedom [13,30]. However, there are known applications of models with a larger number of degrees of freedom [32]. The simulation results of the cutting process are presented in the form of a stability lobes diagram (SLD) used to select the vibration-free machining modes (see, e.g., [25]). Depending on the relative stiffness of the workpiece and cutting tool, three different dynamic models of the technological system (TS) are usually proposed. These models take into account either the tool compliance [30], the workpiece compliance [33,34], or the compliance of the tool–workpiece pair [14]. The authors determined the generalized dynamic characteristics of the tool–workpiece system, only taking into account their coordinate relationship. In this case, the dynamic interaction between the two multiple elastic systems of the machine was not taken into account. Sun and colleagues presented the milling process of a thin-walled plate with an end milling cutter for three dynamic models: an elastic tool, an elastic tool–workpiece pair, and an elastic workpiece model [31]. The model form is determined by the character of the workpiece stiffness change during material removal relative to the tool stiffness. The dynamic model is based only on the vibration responses of the cutter, dynamic parameters, and cutting forces. Similar studies on the stability models for turning cutting processes were considered in [35–37]. Thus, only in dynamic models of a pliable tool–workpiece pair is the cutting process an organic part of the interaction process between the tool and workpiece, which determines the coupling of their vibrations. In other cases, the authors determined the generalized dynamic characteristic of the tool–workpiece system by only taking into account their coordinate coupling, i.e., without taking into account the vibrations coupling.

When using MSTMM to describe the interaction of elastic machine systems through the cutting process, Rui et al. [27] also proposed to consider the contact between the tool and the workpiece in the cutting zone only by a coordinate coupling equation in the form of the joint strain equation in the contact zone. The dynamic vibration coupling of the elements interacting through the cutting process is not taken into account in this case. The kinematics and kinetics consideration principle of mechanics in terms of the MSTMM method was considered by Chen and colleagues in the example of modeling a beam with mountings [26]. They proposed to take into account the coupling between the contacting bodies through the introduction of an additional elastic coupling between them. This is the approach characterized by the receiving communication substructure analysis (RCSA) method. This method was first applied by Schmitz and colleagues to predict tool tip susceptibility during milling using a frequency response function (FRF) [38]. They established the main advantages of the RCSA method [39]. In particular, these advantages are the ability to analytically predict the tool tip FRF, the ability to combine the model and measure the assembly components, and the ability to account for all measured or simulated vibration waveforms within the desired bandwidth without computational cost. Schmitz and Donalson presented an analytical formulation for free tool FRFs when describing the tool as a cantilever beam with distributed mass [40]. Thus, the FRF of individual substructures of the spindle–mandrel–tool system can be calculated analytically or obtained experimentally, and the parameters of their connection are taken into account when obtaining the FRF of the complete assembly [41]. Honeycutt and Schmitz proposed using the RCSA method to predict the frequency response functions when milling a thin-walled plate represented as a beam with varying free end thicknesses [42]. Such an application of the RCSA method to determine the frequency response functions of composite structures allows to extend its application and, also, to describe the dynamic interaction of the workpiece and tool in the cutting process.

## 2. Materials and Methods

The developed model describes the dynamic interaction of the workpiece and tool in the cutting process and ensures the numerical characteristics determination for the coupled vibrations of the machine structure's conjugated elements. The following basic principles were used to develop a dynamic model of a metal-cutting machine tool:

- The interaction of the cutting process with the elastic system of the machine tool is represented by a closed dynamic model;
- The elastic system of a machine tool is a set of coupled mechanical partial subsystems;
- In the closed dynamic model of the machine tool, the cutting process is taken into account as an elastic connection between the subsystems of the workpiece and tool;
- The elastic coupling is given with the stiffness coefficient equal to the ratio of the change in cutting force to the change in depth of cut;
- The coordinate relation between the workpiece and tool subsystems is defined by the condition of joint elastic deformations at the place of their contact. The dynamic vibrations coupling of the workpiece and tool is determined by the stiffness of the elastic coupling between them.

The developed dynamic model was verified by the dynamic identification method [43]. The natural vibration frequencies of a mechanical system are the main dynamic characteristics of this system and provide an opportunity to evaluate the effect of applying dynamic loads of various kinds. Therefore, the experimental verification of the developed dynamic model is carried out through the comparison of the measured and calculated values of natural frequencies of the "spindle–workpiece–tool" subsystem [44]. The workpiece shape, its clamping, and loading parameters were selected so as to ensure a significant effect of these parameters on the change in the natural vibration frequency of the "spindle–workpiece–tool" subsystem. Therefore, solid and tubular workpieces were chosen for the experimental study. Such workpieces differed significantly in mass and insignificantly in stiffness. The different stiffnesses of the workpieces were ensured by the value of their overhang and the point coordinate of the workpiece pressure with the tool [45,46].

### 2.1. Materials

Experimental studies were carried out on Gear Head Lathe GH1230 (Warren Machine Tools Ltd., Guildford, UK, "Warco", 2013 production) in two stages. In the first stage, measurements of the cutting forces during the quasi-orthogonal cutting process were performed. In the second stage, free damping vibrations of the workpiece clamped in the three-cam chuck of the machine were measured. The setup for the experimental studies is shown in Figure 1. Cylindrical rods manufactured from heat-treated AISI 1045 structural steel were used as the workpieces. The mechanical and thermal properties of this material are listed in Table 1.

The solid and tubular workpieces were used for the experiments. The initial diameter of the workpiece was Ø 32 mm, and the inner diameter of the tubular workpiece was Ø 24 ± 0.15 mm. After cutting the workpieces to the required length, they were pre-machined using a longitudinal turning along the outside diameter to the size of Ø 30 mm. The workpiece length during the measurement of the cutting forces and natural frequencies was 180 mm. Before the experimental studies, the workpieces were annealed to ensure a homogeneous material structure and the absence of residual stresses. The hardness of the workpieces after annealing was 180 HB. To measure the free damping vibrations of the workpiece and to implement the orthogonal cutting process, shoulder grooves were formed on the workpiece by plunge turning. Grooving was performed using a cut-off tool with an insert width of 3 mm. The shoulder width was 3 mm (see Figure 1). To implement the process of quasi-orthogonal cutting when measuring cutting forces, a cutting tool equipped with a T15K8 carbide plate from the two-carbide–titanium–tungsten material group of the TC–WC–Co system (material analog: MC111 Sandvik Coromant, Sandviken, Sweden) was used. The T15K8 carbide plate was brazed onto the cutter body. The width of the carbide plate was 5 mm to ensure free orthogonal cutting. The tool rake angle was $\gamma = 0°$, the

clearance angle was $\alpha = 8°$, and the curvature radius of the cutting edge was $\rho = 20$ μm. The cutting speed was $V_C = 160$ m/min, and the cutting feed was $f = 0.2$ mm/rev. When measuring cutting forces, the cutting process was repeated at least 10 times. The results of measuring the cutting forces were averaged over these repetitions. The maximum uncertainty in measuring the cutting forces was not more than 10%.

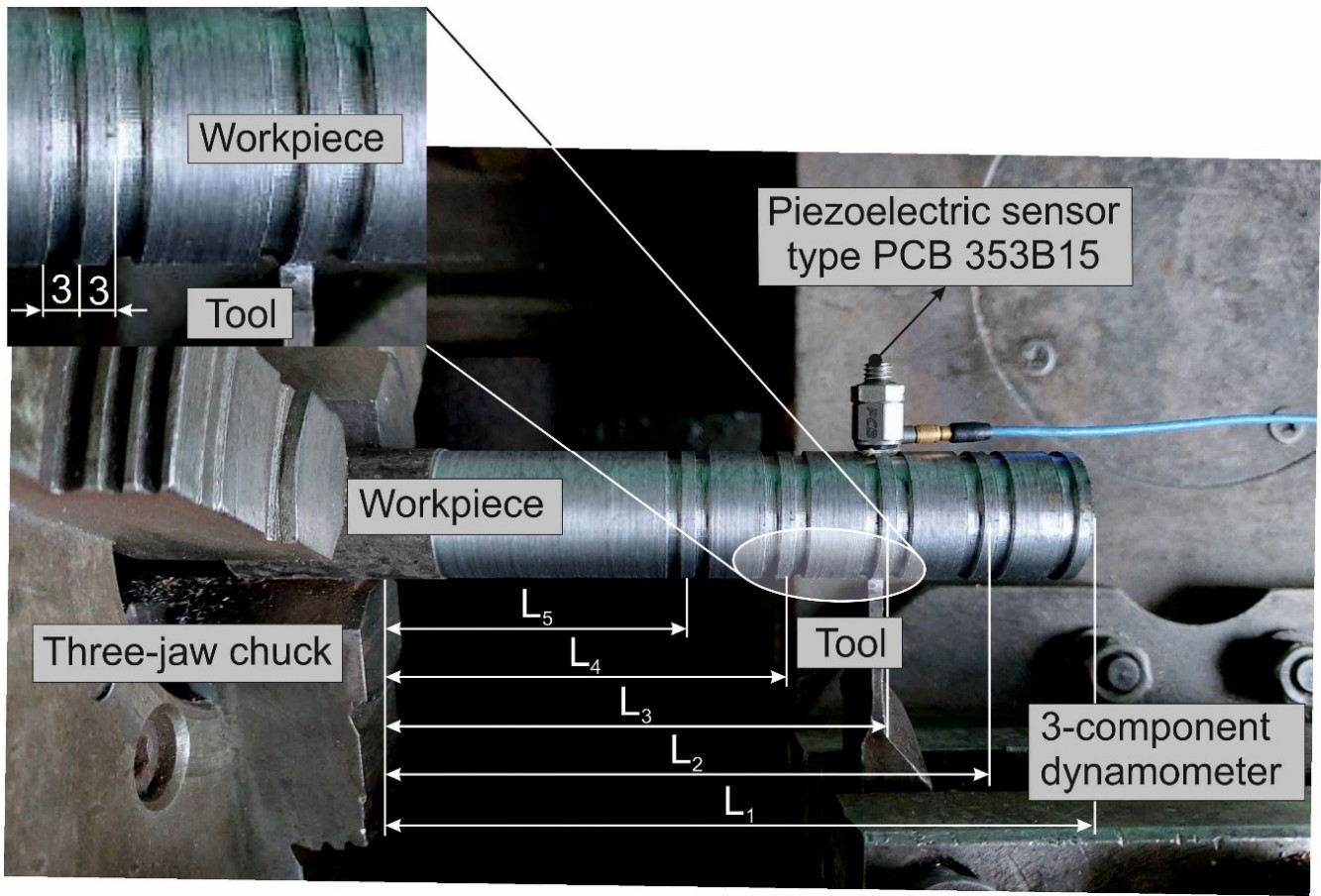

**Figure 1.** Experimental setup for measuring free damped vibrations.

**Table 1.** Mechanical and thermal properties of the steel AISI 1045.

| Strength (MPa) | | Elastic Modulus (GPa) | Elongation (%) | Hardness (HB) | Poisson's Ratio | Specific Heat (J/kg·K) | Thermal Expansion (μm/m · °C) | Thermal Conductivity (W/m·K) |
|---|---|---|---|---|---|---|---|---|
| Tensile | Yield | | | | | | | |
| 690 | 620 | 206 | 12 | 180 | 0.29 | 486 | 14 | 49.8 |

During preliminary workpiece machining along the outside diameter by longitudinal turning, as well as during the formation of shoulder grooves and measuring the cutting forces during free orthogonal cutting, the workpiece was pressed by the machine's tail-stock due to its significant overhang. During the experimental studies to determine the natural frequencies, measurements were carried out on a cantilevered workpiece. This workpiece clamping increased the effect of changing the load application point from the tool. The workpiece shape for implementing the free orthogonal cutting process and for free vibrations measurements was identical.

Before measuring the free damping vibrations, the workpiece was clamped in a three-jaw chuck. The cantilevered clamping of the workpiece enables the best experimental conditions for evaluating the dynamic characteristics of the lathe [47,48]. The length of the console part (overhang) of the workpieces when clamped in the chuck was 122 mm.

Before the measurements, the workpiece was pressed with a cutting tool in the area of the prefabricated shoulder at different distances from the chuck edge to the tool position coordinate. This tool point coordinate varied on five levels in steps of 20 mm, which were $L_1$ = 122 mm, $L_2$ = 102 mm, $L_3$ = 82 mm, $L_4$ = 62 mm, and $L_5$ = 42 mm (see Figure 1). The pressure force of the cutting tool was set by the signal of the dynamometer. The value of the pressure force corresponded to the value of the thrust force $F_T$. This value was set either from the measured values of the cutting forces or according to its value determined using an analytical cutting model or using a numerical simulation. Free damping vibrations were excited by impulse loading of the workpiece through a directed impact with an impact hammer. The vibrations were measured using a PCB 353B15 piezoelectric sensor mounted on the corresponding shoulder of the workpiece. The analog sensor signal was amplified by the PCB 480E09 preamplifier and transferred to the NI USB-9215 measurement card equipped with an analog-to-digital converter. The resulting digital signal was saved and processed on a PC using MATLAB software. A measurement scheme of the workpiece's free damping vibrations is shown in Figure 2.

**Figure 2.** Scheme for measuring the free damped vibrations of the workpiece.

*2.2. Methods*

The model of the spindle unit is developed as part of a closed dynamic machine system. The following assumptions are applied when forming a computational model of a spindle unit using TMM [21]:

- The spindle unit is considered a linear dynamic system with distributed and concentrated parameters;
- The spindle bearings have radial, axial, and angular stiffness, with linear stiffness and damping characteristics;
- The elastic–inertial and damping properties of the spindle and its bearings do not change with the rotation angle, i.e., are isotropic in the plane perpendicular to the spindle rotation axis (axisymmetric problem).

2.2.1. Model of the Spindle Unit System

Taking into account these assumptions, the machine tool spindle in the calculation model is represented in the form of a stepped beam mounted on elastic bearings with viscous damping. Large parts mounted on the spindle (pulleys, gears, etc.) are represented as concentrated masses. The spindle body is divided into several sections (beam elements), separated by an abrupt change in the diameter or by the presence in the spindle section of a concentrated mass, a bearing, or an external concentrated load. Each section is characterized by a constant distributed mass and bending stiffness. The spindle transfer matrix, which relates the stress–strain state parameters at its ends, is the result of the transient matrices product from its sections. The spindle unit dynamic model is represented as an elastic axisymmetric shaft with a distributed mass, modeled by Euler-Bernoulli beam elements. This modeling method is based on the problem solution of natural bending vibrations of a beam element (a beam section) with constant cross-section and a uniformly distributed mass [49]:

$$E \cdot I \cdot \frac{\partial^4 y(x,t)}{\partial x^4} + m \cdot \frac{\partial^2 y(x,t)}{\partial t^2} = 0, \tag{1}$$

where $y(x,t)$—transverse displacement of the beam section; $x$—axial coordinate of the beam section; $E \cdot I$—bending stiffness of the beam section; $m$—unit length mass of the beam section; $t$—time.

Representing the transverse displacement of a beam section with abscissa $x$ as $y(x,t) = y(x) \cdot \sin(\omega \cdot t)$, where $y(x)$ is the amplitude of transverse displacement, and $\omega$ is the circular frequency of the natural vibrations, Equation (1) will be obtained as an ordinary homogeneous differential equation of the fourth order:

$$\frac{\partial^4}{\partial x^4} y(x) + \mu^4 y(x) = 0, \tag{2}$$

where $\mu$ is the frequency parameter of the natural vibrations: $\mu = \sqrt[4]{\frac{m \cdot \omega^2}{E \cdot I}}$.

The solution of Equation (2) will be found in the form:

$$y(x) = y_0 \cdot P(x) + \frac{l}{\mu} \cdot \theta_0 \cdot S(x) + \frac{l^2}{\mu^2 \cdot E \cdot I} \cdot M_0 \cdot R(x) + \frac{l^3}{\mu^3 \cdot E \cdot I} \cdot Q_0 \cdot T(x), \tag{3}$$

where $y_0$, $\theta_0$, $M_0$, and $Q_0$ are the transverse displacement amplitudes, rotation angle, bending moment, and shear force in the initial intersection of the beam section ($x = 0$); $l$ is the length of the beam section; and $P(x)$, $S(x)$, $R(x)$, and $T(x)$ are functions of the forms:

$$P = \tfrac{1}{2} \cdot (\cosh \mu \cdot l + \cos \mu \cdot l); \quad S = \tfrac{1}{2} \cdot (\sinh \mu \cdot l + \sin \mu \cdot l);$$
$$R = \tfrac{1}{2} \cdot (\cosh \mu \cdot l - \cos \mu \cdot l); \quad T = \tfrac{1}{2} \cdot (\sinh \mu \cdot l - \sin \mu \cdot l).$$

Successive differentiations of Equation (3) provide parameters for the rotation angle amplitudes $\theta(x)$, bending moments $M(x)$, and shear forces $Q(x)$ in the section of the beam with abscissa $x$:

$$\theta = \frac{dy}{dx}, \quad M = E \cdot I \cdot \frac{d^2 y}{dx^2}, \quad Q = \frac{dM}{dx} = E \cdot I \cdot \frac{d^3 y}{dx^3},$$

These parameters in the initial section and in the section with abscissa $x$ are related with a vector expression:

$$\{\mathbf{Y}\}_x = [\mathbf{T}_u] \cdot \{\mathbf{Y}\}_0, \tag{4}$$

where $\{\mathbf{Y}\}_0$ and $\{\mathbf{Y}\}_x$ are vectors of parameters in the corresponding sections of the beam section, and $[\mathbf{T}_u]$ is the transfer matrix of a beam element (beam section) with distributed mass [49]:

$$[\mathbf{T}_u] = \begin{bmatrix} P & \frac{S}{\mu} & \frac{R}{\mu^2 \cdot E \cdot I} & \frac{T}{\mu^3 \cdot E \cdot I} \\ \mu \cdot T & P & \frac{S}{\mu \cdot E \cdot I} & \frac{R}{\mu^2 \cdot E \cdot I} \\ \mu^2 \cdot E \cdot I \cdot R & \mu \cdot E \cdot I \cdot T & P & \frac{S}{\mu} \\ \mu^3 \cdot E \cdot I \cdot S & \mu^2 \cdot E \cdot I \cdot R & \mu \cdot T & P \end{bmatrix}, \tag{5}$$

For an unloaded beam consisting of $u$ sections, the matrix equation connecting the parameters at the fore end $\{Y\}_0$ (the 0-th section) and the back end $\{Y\}_u$ (the $u$-th section) of the beam has the form:

$$\{\mathbf{Y}\}_u = [\mathbf{T}] \cdot \{\mathbf{Y}\}_0, \tag{6}$$

where $\{\mathbf{Y}\}_0 = \{y_0, \theta_0, M_0, Q_0\}^T$ and $\{\mathbf{Y}\}_u = \{y_u, \theta_u, M_u, Q_u\}^T$ are vectors of transverse displacement amplitudes $y$, rotation angles $\theta$, bending moments $M$, and shear forces $Q$ at the beam ends, and $[\mathbf{T}]$ is the beam transfer matrix equal to the product of the transfer matrices $[\mathbf{T}]_i$ sections in order from the end of the beam to its beginning.

In general, the transfer matrix is represented by the following expression:

$$[\mathbf{T}] = [\mathbf{T}]_{u,0} = \prod_{i=u}^{0} [\mathbf{T}]_i, \tag{7}$$

In turn, the transfer matrix $[\mathbf{T}]_i$ of each section is composed by taking into account the presence of a concentrated load or bearing in its end section. In general terms, the transfer matrix of the $i$-th section is described by the expression:

$$[\mathbf{T}]_i = [\mathbf{T}_p]_i \cdot [\mathbf{T}_b]_i \cdot [\mathbf{T}_u]_i, \tag{8}$$

where $[\mathbf{T}_p]_i$ is the mass–inertial matrix of a concentrated load; $[\mathbf{T}_b]_i$ is the matrix of an elastic bearing with damping [21,22]:

$$[\mathbf{T}_b] = \begin{bmatrix} 1 & 0 & 0 & 0 \\ 0 & 1 & 0 & 0 \\ 0 & k_a + i \cdot h_a & 1 & 0 \\ -k_r - i \cdot h_r & 0 & 0 & 1 \end{bmatrix}, \tag{9}$$

where $k_r$ and $k_a$—radial and angular bearing stiffnesses; $h_r$ and $h_a$—coefficients characterizing the damping properties of the bearing; $i = \sqrt{-1}$;

$$[\mathbf{T}_p] = \begin{bmatrix} 1 & 0 & 0 & 0 \\ 0 & 1 & 0 & 0 \\ 0 & -J \cdot \omega^2 & 1 & 0 \\ M \cdot \omega^2 & 0 & 0 & 1 \end{bmatrix}, \tag{10}$$

where $M$ is the load mass, and $J$ is the inertia moment of the load relative to the horizontal axis perpendicular to the beam axis.

The transfer matrix $[\mathbf{T}]_0$ of the 0-th section is determined only by the presence of a concentrated load or bearing in the 0-th section: $[\mathbf{T}]_0 = [\mathbf{T}_p]_0 \cdot [\mathbf{T}_b]_0$. In their absence, $[\mathbf{T}]_0 = diag(1,1,1,1)$, and therefore, in determining the transfer matrix $[\mathbf{T}]$, the transfer matrix $[\mathbf{T}]_0$ may not be taken into account.

### 2.2.2. Analytical Method

For the determination of forces in the orthogonal cutting process with the analytical method, the previously developed analytical cutting model was used. To develop a model for determining the kinetic characteristics of the cutting process, the variational principles of plasticity theory—in particular, the principle of minimum potential energy—are

used [50–52]. This principle was taken as a starting point in creating a model of orthogonal cutting with a single shear plane, in which the velocity field of the plastic flow of the machined material is discontinuous. The model was developed for orthogonal cutting of plastic materials, such as structural and alloyed heat-treated steels, as well as alloys, during the machining of which, mainly, the flow chips are formed (Figure 3) [52]. For the application of the minimum energy principle, three energies are considered: chip formation energy in the primary cutting zone—$W_{Ch}$, friction energy in the secondary cutting zone—$W_{FS}$, and friction energy in the tertiary cutting zone—$W_{FT}$.

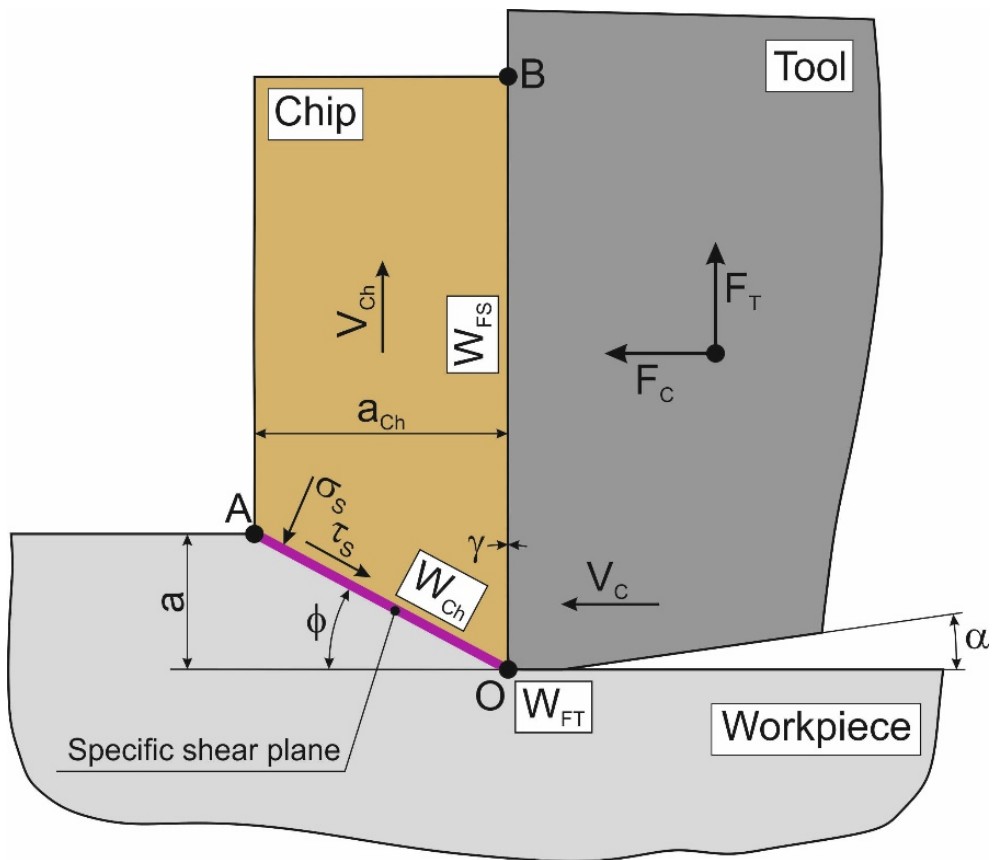

**Figure 3.** The layout of chip formation for the orthogonal cutting process. The following symbols are used in the figure: $V_C$—cutting speed; $V_{Ch}$—chip speed; $a_{Ch}$—chip thickness; $\sigma_S$—normal stress,.

By using the applied variational principle, the sum of the considered energies $W_C$ should tend toward a minimum. Thus, the following cutting power balance equation can be postulated [52]:

$$\underset{\Psi}{\forall}\{P\} \underset{k=1}{\overset{n}{\exists}}\{W_k \in P\} :\Rightarrow W_C = W_{Ch} + W_{FS} + W_{FT} = \sum_{k=1}^{n} W_k \Rightarrow 0, \qquad (11)$$

where $\Psi$—is the existence space of the cutting process states (conditions); $P$—is the cutting process state; $k$ and $n$—are indices.

Expanding Equation (11) by determining the specified energies according to the considered cutting characteristics, the contact characteristics of the tool with the machined material, and the tool geometry parameters and performing the necessary transformations

with the variables substituted provides an equation system for determining the cutting force and thrust force during orthogonal cutting:

$$\begin{cases} F_C = \frac{\tau_S \cdot a \cdot w}{\cos(\phi-\gamma) - f_S \cdot \sin(\phi-\gamma)} \cdot (\cos\gamma + f_T \cdot \sin\gamma) + q_F \cdot h_1 \cdot w \cdot f_T \\ \\ F_T = \frac{\tau_S \cdot a \cdot w}{\cos(\phi-\gamma) - f_S \cdot \sin(\phi-\gamma)} \cdot (\sin\gamma + f_T \cdot \cos\gamma) + q_F \cdot h_1 \cdot w \end{cases}, \qquad (12)$$

where $F_C$ and $F_T$ are the cutting and thrust force *s* accordingly; $\tau_S$ is the shear yield point of the machined material; $a$ is the undeformed chip thickness (depth of cut for the orthogonal cutting process); $w$ is the cutting width; $\phi$ is the shear angle; $\gamma$ is the tool rake angle; $q_F$ is the contact pressure at the clearance face; $f_S$ is the friction coefficient at the rake face; $f_T$ is the friction coefficient at the clearance face; $h_1$ is the length of chamfer wear at the clearance face.

The determination of the contact pressure at the clearance face $q_F$ was due to the solution of Prandtl's problem, adapted for orthogonal machining [52,53]. The shear angle $\phi$ for the respective cutting and material parameters, as well as for the tool geometry parameters, is calculated based on the variational principle of the minimum cutting power [52]. To determine the shear yield point of the material to be machined $\tau_S$, studies of the resistance of the processed material to the cutting process were used [53]. The friction coefficient $f_S$ in the contact between the tool rake face and the chip, as well as the friction coefficient $f_T$ in the contact between the tool clearance face and the machined workpiece, were determined for the specified experimental conditions per the procedures [54,55].

### 2.2.3. FEM

The numerical simulation of the orthogonal cutting process to determine the cutting forces was performed using a finite element cutting model. For this purpose, a three-dimensional cutting model was developed to simulate the machining of AISI 1045 steel. Figure 4 shows a meshed geometric model of the developed FEM model, combined with the simulation results of the machined material strain. In addition, Figure 4 also shows the initial and boundary conditions of the model.

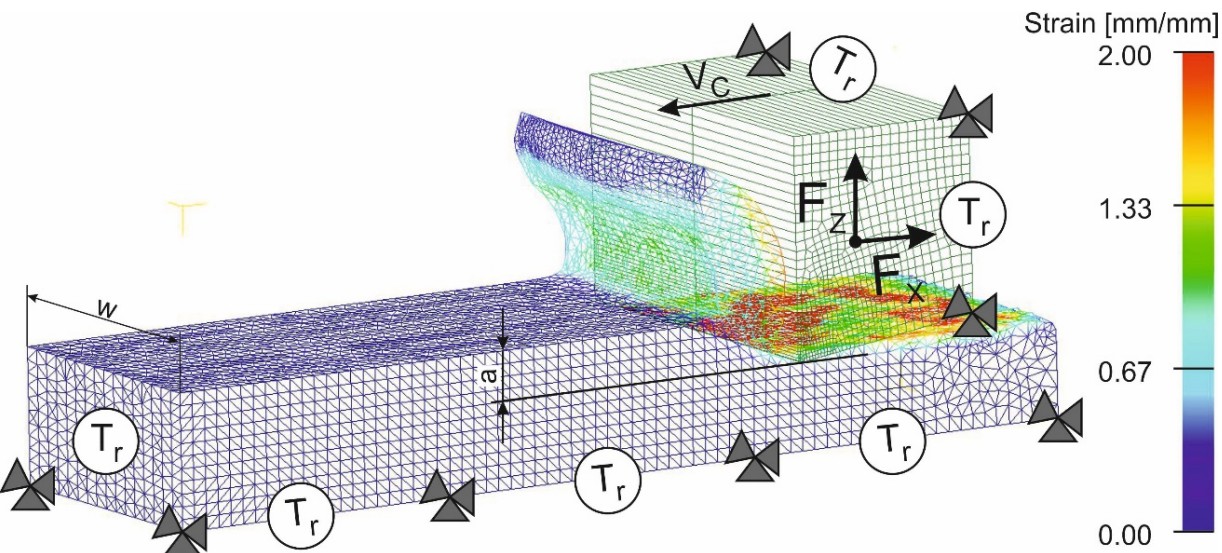

**Figure 4.** Initial geometry and boundary conditions of the FEM cutting model.

In the primary cutting zone, as well as in the zone of contact between the tool rake face and the contact zone between the tool clearance face and the workpiece machined surface, the mesh was made significantly finer than in the rest of the workpiece. The mesh in the remaining areas was established more roughly. This was done to improve the efficiency

and accuracy of the cutting process simulation. Through the fixing of the workpiece and tool movements in the direction of the coordinate axes, as well as by setting the thermal conditions at the boundaries of the tool and workpiece, the boundary conditions were determined. The tool movement in the Z-axis direction was limited by rigidly fixing its surface opposite to the tool rake face. The workpiece movement was limited by the rigid fixation of its bottom in the directions of the X, Y, and Z axes. The thermal initial conditions at room temperature ($T_r$) were given at the bottom and the left-hand side of the workpiece, as well as at the left-hand side and the top of the tool. The tool penetrated the workpiece along axis Z to the depth of cut $a$ (the thickness of the undeformed chip). The workpiece width is given by the value $w$. The tool working motion with the cutting speed $V_C$ was set to an absolute motion in the negative X-direction to ensure the cutting process. The initial workpiece model mesh contained about 58,237 elements and about 12,901 nodes. The edge length of the largest workpiece model element was about 0.096 mm, and the edge length of the smallest element was about 0.029 mm. The initial tool model mesh contained about 28,668 elements and about 6498 nodes. In this case, the edge length of the largest element was about 0.093 mm, and the edge length of the smallest element was about 0.0196 mm.

The material model of AISI 1045 steel was described by the Johnson–Cook constitutive equation [56] with the model parameter values presented in [57,58]. The contact interaction between the tool rake face and the chip in the secondary cutting zone, as well as the contact interaction between the tool clearance face and the workpiece, were modeled using the Coulomb model in accordance with the method [54,55]. A special model of machined material damage [59], such as when machining titanium alloys [60], was not provided in the developed FE-cutting model. This was done because chip formation when machining AISI 1045 steel is a continuous process with the formation of flow chips [61,62]. In this case, the machined material damage that realized the chip separation process occurred automatically, according to the algorithm used in the software package [63].

## 3. Results

The application of RCSA to describe the dynamic interaction between the cutting process and machine structure was formulated by Danylchenko and colleagues [64]. They proposed to consider the cutting process as a dynamic interaction between the tool and the workpiece. The tool and the workpiece are represented as elastic bodies with a distributed mass, and the cutting process is represented as an elastic connection between them in the cutting zone. The stiffness of this elastic coupling $k_p$ was taken to be equal to the ratio of the change in cutting force $F_T$ to the change in cutting depth $a_p$ [65]. This relation can also be represented in the form of a partial derivative of the function $F_T = F_T(a_p)$, namely $k_p = \frac{\partial F_T}{\partial a_p}$. In the linear cutting force model, the stiffness $k_p$ corresponds to the static stiffness of the cutting force equal to the product of the cutting force coefficient $k_f$ and the chip width $w$ [30,34]. The representation of the cutting process in the form of an elastic connection between the workpiece and the tool allows taking into account not only the coordinate connection between them in the contact zone [27] but also the dynamic connectivity of their vibrations [66,67]. This, in turn, makes it possible to consider the cutting process as a dynamic component of the mechanical system "spindle unit–workpiece–tool" and to present the system itself as a closed dynamic system [68].

### 3.1. Dynamic Model of System "Spindle Unit"

The dynamic model of the spindle is developed as a model part of the lathe's closed dynamic system. The machine model is represented by a set of dynamic models of the elastic system "spindle unit" and the tool elastic system. These systems interact with each other through the cutting process. A dynamic model of the elastic spindle unit system is developed for a spindle unit with a workpiece clamped in the spindle. A dynamic model of the elastic tool system is developed for a tool mounted on the machine bed. The workpiece interacts with the tool through the cutting process. In open-loop dynamic models of elastic systems, the cutting process is taken into account only by the cutting force [8].

The spindle unit is represented as a subsystem set of a mechanical oscillating system. The "spindle unit" system includes subsystem 1 of the workpiece (index $s = 1$), subsystem 2 of the spindle (index $s = 2$), and subsystem 3 of the spindle housing (index $s = 3$) (Figure 5). The elastic tool system is represented as a concentrated mass $m_t$ elastically fixed to the machine bed. For the "spindle unit" system, the connections between its subsystems are the connection between the workpiece and spindle for subsystems 1 and 2, the spindle bearing for subsystems 2 and 3, and the connection between the spindle housing and the machine bed for subsystem 3. All connections have elastic and dissipative properties. During machining, the "spindle unit" system interacts with the tool system 0 (index $s = 0$). In the dynamic model, this interaction is accounted for by the elastic connection between stiffness $k_p^{(1)}$ and force $\widetilde{F}_1^{(1)}$. The normal for the cutting plane component of the cutting force is used as the force (thrust force). In the general case, this force $\widetilde{F}_1^{(1)}$ is a variable that is functionally dependent on the actual depth of cut $a$. The force $F_s^{(1)}$ has a static component and a dynamic component $F_d^{(1)}$ [30]. The static component $F_s^{(1)}$ is determined by a given value of the cutting depth and the value of the workpiece static displacement relative to the tool at the place of its application. The variable component $F_d^{(1)}$ is determined by periodic changes in the cutting depth. The depth variation is caused by the variable chip thickness being cut and the value of the workpiece dynamic displacement relative to the tool at the point where the force is applied.

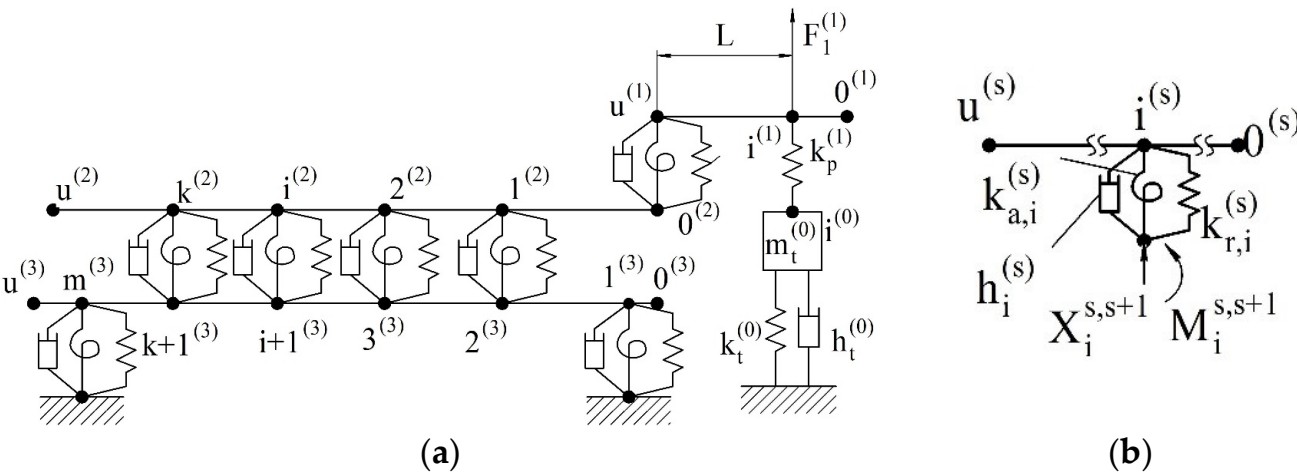

**Figure 5.** A model of the lathe's closed dynamic system: (**a**) scheme of the closed dynamic model; (**b**) generalized scheme of the supports (joints).

Taking into account the back-coupling between the elastic system and the cutting process, the static component of the workpiece and tool relative displacement and the static component of the actual depth of the cut are determined from the force $F_s^{(1)}$. The force value $F_d^{(1)}$ is used to determine the dynamic component of the workpiece and tool relative displacement and the dynamic component of the actual depth of the cut.

In the dynamic behavior study of complex mechanical systems, it is considered reasonable to divide them (decompose) into simpler subsystems [26]. The separation of subsystems from each other is ensured either by introducing additional connections that prohibit the common points (nodes) movement of subsystems (method of dynamic stiffnesses) or, in contrast, by eliminating the connections between them (method of dynamic compliances). When decomposing the system using the method of dynamic compliances, their harmonic reactions are applied in the direction of the removed connections. These reactions are then determined from the condition of joint deformation of the subsystems [41,69]. Applying this approach to the dynamic model of the "spindle assembly" system, it is represented as a set of interconnected dynamic models of subsystems under the harmonic force action.

The dynamic models of the subsystems extracted from the "spindle unit" system by replacing the connections between them with appropriate harmonic reactions are shown in Figure 6. Workpiece subsystem 1 is illustrated in Figure 6a. Tool subsystem 0 is illustrated in Figure 6b. Spindle subsystem 2 is shown in Figure 6c. Housing subsystem 3 is shown in Figure 6d.

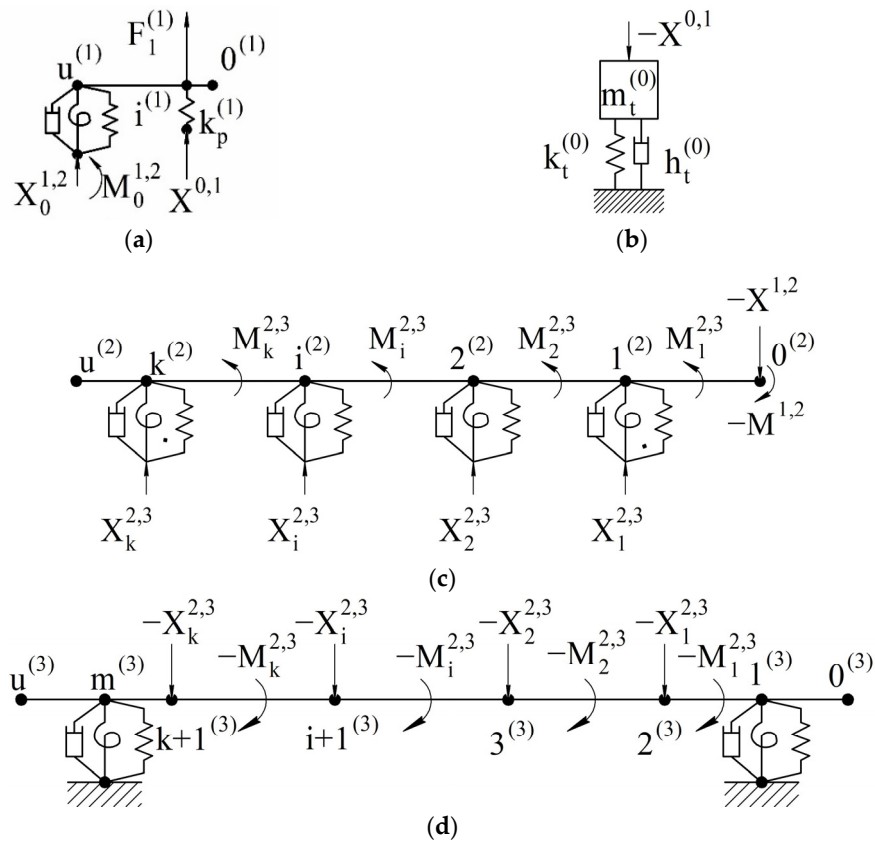

**Figure 6.** Subsystem dynamic models of the "spindle unit" system: (**a**) 1—workpiece; (**b**) 0—tool; (**c**) 2—spindle; (**d**) 3—housing.

The conditions for subsystem deformation jointness in this case are represented in the form of equality conditions for the amplitudes of generalized displacements (linear and angular) at the points of subsystem separation. At points $u^{(1)}$ (see Figure 6a) and $0^{(2)}$ (see Figure 6c) of subsystems 1 and 2 separation, these conditions have the form:

$$
\begin{cases}
\alpha_{uu}^{(1,2)} \cdot X^{1,2} + \gamma_{uu}^{(1)} \cdot M^{1,2} + \alpha_{u1}^{(1)} \cdot X^{0,1} + \alpha_{uF}^{(1)} = -\alpha_{00}^{(2)} \cdot X^{1,2} - \gamma_{00}^{(2)} \cdot M^{1,2} + \sum_{j=1}^{k} \alpha_{0j}^{(2)} \cdot X_j^{2,3} + \\
+ \sum_{j=1}^{k} \gamma_{0j}^{(2)} \cdot M_j^{2,3} \\
\beta_{uu}^{(1)} \cdot X^{1,2} + \phi_{uu}^{(1,2)} \cdot M^{1,2} + \beta_{u1}^{(1)} \cdot X^{0,1} + \beta_{uF}^{(1)} = -\beta_{00}^{(2)} \cdot X^{1,2} - \phi_{00}^{(2)} \cdot M^{1,2} + \sum_{j=1}^{k} \beta_{0j}^{(2)} \cdot X_j^{2,3} + \\
+ \sum_{j=1}^{k} \phi_{0j}^{(2)} \cdot M_j^{2,3}
\end{cases}
\tag{13}
$$

where $X^{0,1}$, $X^{1,2}$, $X_j^{2,3}$, $M^{1,2}$, and $M_j^{2,3}$ are the amplitudes of the removed connection reactions (forces and moments); $\alpha_{ij}^{(s)}$, $\beta_{ij}^{(s)}$, $\gamma_{ij}^{(s)}$, and $\phi_{ij}^{(s)}$ are the harmonic coefficients of the subsystems' $s$ influence, namely $\alpha_{ij}^{(s)}$ and $\beta_{ij}^{(s)}$ as the amplitudes of the displacement and rotation angle in the $i$-th point from the unit harmonic force applied in the $j$-th point; $\gamma_{ij}^{(s)}$

and $\phi_{ij}^{(s)}$ are the amplitudes of the displacement and angle of rotation at the $i$-th point from the unit harmonic moment applied at the $j$-th point; $\alpha_{uu}^{1,2}$ and $\phi_{uu}^{1,2}$ are the harmonic influence coefficients at the points of subsystems 1 and 2 separation; $\alpha_{uF}^{(1)}$ and $\beta_{uF}^{(1)}$ are the amplitudes of generalized displacements (linear and angular) at the $u$ point of subsystem 1 from the applied external harmonic force.

The harmonic influence coefficients $\alpha_{uu}^{1,2}$ and $\phi_{uu}^{1,2}$ take into account the compliances of the removed connections and, by analogy with [69], are determined by dependencies:

$$\alpha_{uu}^{1,2} = \alpha_{uu}^{(1)} + \frac{1}{k_r^{(1)}} \text{ and } \phi_{uu}^{1,2} = \phi_{uu}^{(1)} + \frac{1}{k_a^{(1)}}, \tag{14}$$

where $k_r^{(1)}$ and $k_a^{(1)}$ are the radial and angular stiffnesses of connection of subsystems 1 and 2.

The amplitudes of the generalized displacements $\alpha_{uF}^{(1)}$ and $\beta_{uF}^{(1)}$ from the applied external harmonic force are determined by the dependencies:

$$\alpha_{uF}^{(1)} = \alpha_{u1}^{(1)} \cdot F_1^{(1)}; \; \beta_{uF}^{(1)} = \beta_{u1}^{(1)} \cdot F_1^{(1)}, \tag{15}$$

The deformation coincidence conditions for subsystems 2 (see Figure 6c) and 3 (see Figure 6d) at the $i$-th separation point are represented as:

$$\begin{cases} -\alpha_{i0}^{(2)} \cdot X^{1,2} - \gamma_{i0}^{(2)} \cdot M^{1,2} + \sum\limits_{j=1, \, j \neq i}^{k} \alpha_{ij}^{(2)} \cdot X_j^{2,3} + (\alpha_{ii}^{(2)} + \frac{1}{k_{r,i}^{(2)}}) \cdot X_i^{2,3} + \sum\limits_{j=1}^{k} \gamma_{ij}^{(2)} \cdot M_j^{2,3} = \\ = -\sum\limits_{j=1}^{k} \alpha_{ij}^{(3)} \cdot X_j^{2,3} - \sum\limits_{j=1}^{k} \gamma_{ij}^{(3)} \cdot M_j^{2,3} \\ -\beta_{i0}^{(2)} \cdot X^{1,2} - \phi_{i0}^{(2)} \cdot M^{1,2} + \sum\limits_{j=1}^{k} \beta_{ij}^{(2)} \cdot X_j^{2,3} + \sum\limits_{j=1, \, j \neq i}^{k} \phi_{ij}^{(2)} \cdot M_j^{2,3} + (\phi_{ii}^{(2)} + \frac{1}{k_{a,i}^{(2)}}) \cdot M_i^{2,3} = \\ = -\sum\limits_{j=1}^{k} \beta_{ij}^{(3)} \cdot X_j^{2,3} - \sum\limits_{j=1}^{k} \phi_{ij}^{(3)} \cdot M_j^{2,3} \end{cases} \tag{16}$$

where $k_{r,i}^{(2)}$ and $k_{a,i}^{(2)}$ are the radial and angular stiffnesses of the $i$-th spindle support.

The deformation coincidence conditions for subsystems 1 (see Figure 6a) and 0 (see Figure 6b), taking into account the stiffness $k_p^{(1)}$ of the cutting process, are represented as follows:

$$(\alpha_{11}^{(1)} + \frac{1}{k_p^{(1)}}) \cdot (F_1^{(1)} + X^{0,1}) + \alpha_{1u}^{(1)} \cdot X^{1,2} + \gamma_{1u}^{(1)} \cdot M^{1,2} = -\alpha_t^{(0)} \cdot X^{0,1}, \tag{17}$$

where $\alpha_t^{(0)}$ is the harmonic coefficient of influence of the subsystem 0-th.

The full system of deformation jointness equations in matrix form is described by the following equation:

$$\begin{bmatrix} \alpha(\omega) & \gamma(\omega) \\ \beta(\omega) & \phi(\omega) \end{bmatrix} \cdot \begin{bmatrix} \mathbf{X} \\ \mathbf{M} \end{bmatrix} = \begin{bmatrix} \alpha_{\mathbf{F}} \\ \beta_{\mathbf{F}} \end{bmatrix} \text{ or } [\mathbf{D}(\omega)] \cdot [\mathbf{F}] = [\Delta_{\mathbf{F}}], \tag{18}$$

where $[\mathbf{D}(\omega)]$, $[\mathbf{F}]$, and $[\Delta_{\mathbf{F}}]$ are the block matrices of dynamic compliance (harmonic influence coefficients), amplitudes of the generalized reactions of removed connections, and generalized displacements from the applied external harmonic force influences.

Equations system (16) includes equations system (11), "$k$" equations systems (16) (according to the number of connections in systems 2 and 3), and Equation (17). The separate components of these equations are determined by dependencies (14) and (15). From equations system (17), the natural frequencies vibration of the elastic system, the reactions of the removed connections [**X**] and [**M**], and the transverse (radial) displacement amplitudes

$q_i^{(s)}$ of the *i*-th points of the subsystems are determined. The movement amplitudes $q_i^{(1)}$ of the workpiece subsystem 1 are determined using the dependencies:

$$q_i^{(1)} = \alpha_{i1}^{(1)} \cdot (F_1^{(1)} + X^{0,1}) + \alpha_{iu}^{(1)} \cdot X^{1,2} + \gamma_{iu}^{(1)} \cdot M^{1,2}, \tag{19}$$

The movement amplitudes $q_i^{(2)}$ of the spindle subsystem 2 are defined using the dependencies:

$$q_i^{(2)} = -\alpha_{i0}^{(2)} \cdot X^{1,2} - \gamma_{i0}^{(2)} \cdot M^{1,2} + \sum_{j=1,\, j \neq i}^{k} \alpha_{ij}^{(2)} \cdot X_j^{2,3} + (\alpha_{ii}^{(2)} + \frac{1}{k_{r,i}^{(2)}}) \cdot X_i^{2,3} + \sum_{j=1}^{k} \gamma_{ij}^{(2)} \cdot M_j^{2,3}, \tag{20}$$

The movement amplitudes $q_i^{(3)}$ for subsystems 3 housing are defined using the dependencies:

$$q_i^{(3)} = -\sum_{j=1}^{k} \alpha_{ij}^{(3)} \cdot X_j^{23} - \sum_{j=1}^{k} \gamma_{ij}^{(3)} \cdot M_j^{23}. \tag{21}$$

The displacement amplitudes of the tool subsystem 0 are determined using the following dependencies:

- under the force $F_1^{(1)}$ action (open-loop system):

$$q^{(0)} = -\alpha_t^{(0)} \cdot F_1^{(1)}; \tag{22}$$

- when representing the cutting process by taking into account the elastic relationship with stiffness $k_p^{(1)}$ (closed system):

$$q^{(0)} = -\alpha_t^{(0)} \cdot X^{0,1}, \tag{23}$$

The basic transfer matrices (5), (9), and (10) are used to calculate the harmonic influence coefficients in the joint deformation equations (18). To derive the calculated dependencies of the harmonic influence coefficients in the joint deformation equations of subsystem (18), it is necessary to consider two special cases of loading a beam with a distributed mass mounted on elastic bearings with damping (Figure 7):

- loading by harmonic force $F_j$ (Figure 7a);
- loading by harmonic moment $M_j$ (Figure 7b).

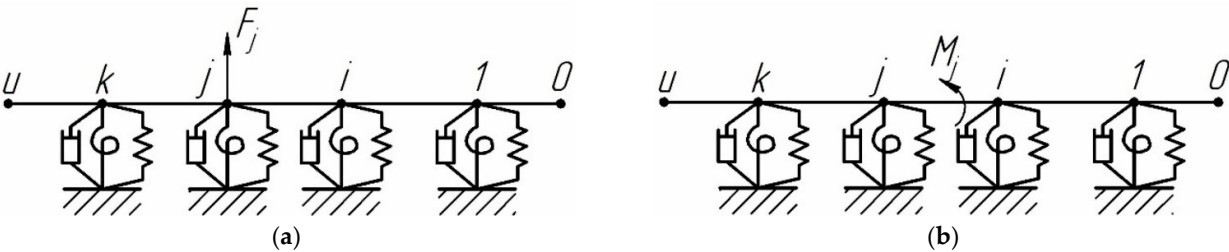

(a)          (b)

**Figure 7.** Schemes for determining the harmonic influence coefficients of the subsystems: (**a**) loading by harmonic force $F_j$; (**b**) loading by harmonic moment $M_j$.

For the beam shown in Figure 7, the parameter vectors $\{Y\}_0$ and $\{Y\}_u$, as well as the transfer matrix $[T]$, will be expressed using the following equations (see Section 2.1):

$$\{Y\}_0 = \{y_0, \theta_0, 0, 0\}^T; \quad \{Y\}_u = \{y_u, \theta_u, 0, 0\}^T; \quad [T] = [T]_u \cdot [T]_k \cdot [T]_j \cdot [T]_i \cdot [T]_1 = \prod_{i=u}^{1} [T]_i, \tag{24}$$

Taking into account Equation (8), the transfer matrix $[\mathbf{T}]$ of the beam in the expanded form is:

$$[\mathbf{T}] = [\mathbf{T}_u]_u \cdot [\mathbf{T}_b]_k \cdot [\mathbf{T}_u]_k \cdot [\mathbf{T}_b]_j \cdot [\mathbf{T}_u]_j \cdot [\mathbf{T}_b]_i \cdot [\mathbf{T}_u]_i \cdot [\mathbf{T}_b]_1 \cdot [\mathbf{T}_u]_1, \tag{25}$$

In the case of an external load applied to the *j*-th cross-section, Equation (6) for the schemes shown in Figure 7 appears as follows [21]:

$$\{\mathbf{Y}\}_u = [\mathbf{T}] \cdot \{\mathbf{Y}\}_0 + [\mathbf{T}]_{u,j} \cdot \{\mathbf{F}\}_j, \tag{26}$$

where $[\mathbf{F}]_j$ is the load vector in the *j*-th section of the beam: $\{\mathbf{F}\}_j = \{0,0,0,F_j\}^T$ or $\{\mathbf{F}\}_j = \{0,0,M_j,0\}^T$; $[\mathbf{T}]_{u,j}$ is the matrix equal to the product of the transfer matrices $[\mathbf{T}]_i$ of the sections (8) placed between the *u*-th and *j*-th sections:

$$\{\mathbf{Y}\}_i = [\mathbf{T}]_{i,0} \cdot \{\mathbf{Y}\}_0 + [\mathbf{T}]_{i,j} \cdot \{\mathbf{F}\}_j, \tag{27}$$

where $\{\mathbf{Y}\}_i = \{y_i, \theta_i, M_i, Q_i\}^T$ is the vector of the transverse displacement amplitudes $y_i$, rotation angle $\theta_i$, bending moment $M_i$, and shear force $Q_i$ in the *i*-th beam section; $[\mathbf{T}]_{i,0}$ is a matrix equal to the product of the transfer matrices $[\mathbf{T}]_i$ of the sections (8) placed between the *i*-th and 0-th sections; $[\mathbf{T}]_{i,j}$ is a matrix equal to the product of the transfer matrices $[\mathbf{T}]_i$ of the sections (8) placed between the *i*-th and *j*-th sections.

The order in which the $[\mathbf{T}]_{u,j}$, $[\mathbf{T}]_{i,0}$, and $[\mathbf{T}]_{i,j}$ matrices included in Equations (26) and (27) and their corresponding harmonic influence coefficients are determined is presented in Appendix A.

### 3.2. Stiffness Calculation of the Additional Elastic Coupling

The stiffness $k_p^{(1)}$ of the additional elastic coupling is taken into account in the joint strain in Equation (17) and, accordingly, is included in the dynamic compliance matrix $[\mathbf{D}(\omega)]$ (18). In the general case, the natural frequencies of dynamic systems are determined from the condition det $[\mathbf{D}(\omega)] = 0$. Therefore, the influence of the value $k_p^{(1)}$ on the natural frequencies vibration formation of the considered closed dynamic system of the lathe is obvious. In general terms, this stiffness is the ratio of the thrust component of the cutting force $F_T$ to the change in the depth of cut (undeformed chip thickness) [34] and is defined by the dependence [64]:

$$k_p = \frac{\partial F_T}{\partial a_{pr}}, \tag{28}$$

where $a_{pr}$ is the actual value of the depth of the cut.

Various methods can be used to determine the thrust force $F_T$: direct measurement of cutting forces in the studied cutting process (see Section 2.1), the use of various known empirical dependencies, calculation of the cutting forces by the analytical cutting model (see Section 2.2.2), and finally, by numerical simulation of the cutting process (see Section 2.2.3). For example, a well-known empirical relationship [70] can be used to determine the thrust force $F_T$ in the turning process:

$$F_T = C_p \cdot a_{pr}^x \cdot f^y \cdot V_C^n \cdot k. \tag{29}$$

The value $k_p^{(1)}$ is determined as follows:

$$k_p^{(1)} = \frac{\partial F_T}{\partial a_{pr}} = \frac{F_T}{a_{pr}} = C_p \cdot x \cdot a_{pr}^{x-1} \cdot f^y \cdot V_C^n \cdot k, \tag{30}$$

where $C_p$—coefficient that depends on the mechanical properties and structure of the machined material and the material of the cutter cutting part, as well as the processing type; $f$—feed; $V_C$—cutting speed; $x$, $y$, and $n$—degree values; $k$—correction factor that takes into account the actual cutting conditions.

In general, the actual value of the cutting depth $a_{pr}$ depends on the total elastic displacements of the workpiece $\widetilde{q}_1^{(1)}$ and tool $\widetilde{q}_c^{(0)}$ at contact point $1^{(1)}$.

They are determined by static and dynamic (dependencies (19) and (23)) calculations:

$$a_{pr} = a_p - \widetilde{q}_t^{(0)} - \widetilde{q}_1^{(1)}, \tag{31}$$

Taking into account dependencies Equations (19) and (23), $a_{pr}$ is defined using the dependence:

$$a_{pr} = a_p + \widetilde{\alpha}_t^{(0)} \cdot \widetilde{X}^{0,1} - \widetilde{\alpha}_{i1}^{(1)} \cdot \widetilde{X}^{0,1} - \widetilde{\alpha}_{i2}^{(1)} \cdot \widetilde{X}^{1,2} - \widetilde{\gamma}_{i2}^{(1)} \cdot \widetilde{M}^{1,2}, \tag{32}$$

It is obvious that a change in the actual value of the cutting depth $a_{pr}$ leads to a change in the value of the radial component of the cutting force $F_T$ (29), the value of the stiffness $k_p^{(1)}$ (30), and the values of $\widetilde{X}^{0,1}$, $\widetilde{X}^{1,2}$, and $\widetilde{M}^{1,2}$, the reactions of the removed connections (17). Thus, the model takes into account the closed elastic system "spindle–workpiece–tool" during cutting. In the case of a linear cutting force characteristic, the stiffness $k_p^{(1)}$ can be determined by using the empirical dependence (29):

$$k_p^{(1)} = F_T / a_p = C_p \cdot x \cdot a_p^{x-1} \cdot f^y \cdot V_C^n \cdot k, \tag{33}$$

where $a_p$ is the given value of the cutting depth.

### 3.3. Determination of Natural Frequencies Vibration of the Lathe Elastic System "Spindle–Workpiece–Tool"

3.3.1. Dynamic Model of the Lathe "Spindle–Workpiece–Tool" System

The studied spindle unit is mounted on a solid body rigidly fixed to the machine bed, so its influence on the formation of natural vibration frequencies of the spindle unit can be neglected. Based on this, the spindle unit is considered a mechanical vibration system consisting of the subsystems of the workpiece (subsystem 1, index $s = 1$) and the spindle with the chuck (subsystem 2, index $s = 2$), which is elastically fixed to the machine bed. The "spindle unit" system interacts with the tool subsystem (subsystem 0, index $s = 0$) at the point of contact between the workpiece and the tool. The structural and calculation scheme of the spindle unit, taking into account the contact interaction between the workpiece and tool, is shown in Figure 8.

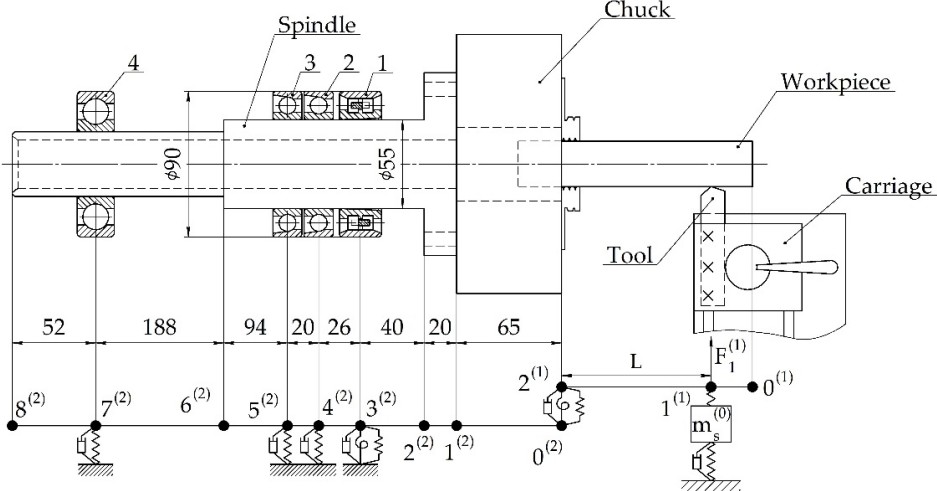

**Figure 8.** Structural and calculation scheme of the spindle unit.

The workpiece subsystem ($s = 1$) is considered as a beam with distributed mass consisting of two sections ($u = 2$). In section $1^{(1)}$, the workpiece is in contact with the

tool. This is taken into account by the elastic coupling with stiffness $k_p^{(1)}$. The preload of this elastic connection during cutting is provided by moving the slide with the tool to a given depth of cut $a_p$. When the experimental studies are performed, this preload is ensured by pressing the workpiece with a force equivalent to the thrust force $F_T$. In section $2^{(1)}$, the workpiece is connected to the spindle (section $0^{(2)}$ spindle). This is taken into account via elastic couplings with stiffnesses $k_r^{(1)}$ and $k_a^{(1)}$ and damping coefficient $h^{(1)}$. The spindle–chuck subsystem ($s = 2$) is considered as an elastically mounted beam with a distributed mass on the machine bed. The beam consists of eight sections ($u = 8$). In sections $3^{(2)}$, $4^{(2)}$, $5^{(2)}$, and $7^{(2)}$ of the beam, bearings are placed. This is taken into account by the elastic and damping connections with radial $k_{r,i}^{(2)}$ and angular $k_{a,i}^{(2)}$ stiffnesses, as well as the damping coefficients $h_i^{(2)}$, ($i = 1$–$4$). The tool subsystem ($s = 0$) is considered as a concentrated mass $m_t^{(0)}$ elastically mounted on the machine's slide. This is taken into account by the stiffness $k_t^{(0)}$ and damping $h_t^{(0)}$ coefficients, respectively. Thus, the "spindle unit" system appears as a "spindle–workpiece–tool" system. To mathematically describe the dynamic behavior of the "spindle–workpiece–tool" system (see Figure 8), this system is decomposed [71]. Subsystems are separated from the system by replacing the connections between them with appropriate harmonic reactions. The system decomposition scheme when taking into account the contact interaction between the workpiece and tool is shown in Figure 9.

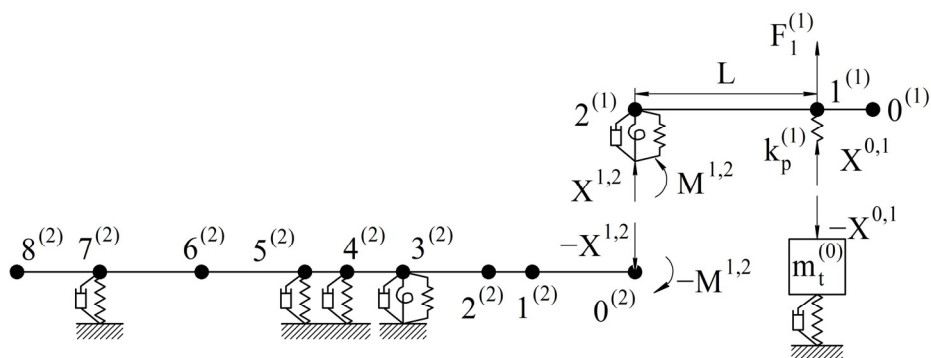

**Figure 9.** Decomposition scheme of the "spindle–workpiece–tool" system during the contact interaction of the workpiece with the tool: index (1)—workpiece subsystem; index (2)—spindle–tool–chuck subsystem; index (0)—tool subsystem.

The dynamic behavior of the system is described by the joint deformation conditions of the subsystems. This deformation jointness of the subsystems is represented in the form of equality conditions for the amplitudes of linear displacements at the points of subsystem separation (see Figure 9):

- for the subsystems of the workpiece ($s = 1$) and tool ($s = 0$):

$$(\alpha_{11}^{(1)} + \frac{1}{k_p^{(1)}}) \cdot X^{0,1} + \alpha_{11}^{(1)} \cdot F_1^{(1)} + \alpha_{12}^{(1)} \cdot X^{1,2} + \gamma_{12}^{(1)} \cdot M^{1,2} = -\alpha_t^{(0)} \cdot X^{0,1}, \qquad (34)$$

➢ for the subsystems of the workpiece ($s = 1$) and the spindle with chuck ($s = 2$):4

$$\begin{cases} \alpha_{21}^{(1)} \cdot X^{0,1} + \alpha_{21}^{(1)} \cdot F_1^{(1)} + (\alpha_{22}^{(1)} + \frac{1}{k_r^{(1)}}) \cdot X^{1,2} + \gamma_{22}^{(1)} \cdot M^{1,2} = -\alpha_{00}^{(2)} \cdot X^{1,2} - \gamma_{00}^{(2)} \cdot M^{1,2} \\ \beta_{21}^{(1)} \cdot X^{0,1} + \beta_{21}^{(1)} \cdot F_1^{(1)} + \beta_{22}^{(1)} \cdot X^{1,2} + (\phi_{22}^{(1)} + \frac{1}{k_a^{(1)}}) \cdot M^{1,2} = -\beta_{00}^{(2)} \cdot X^{1,2} - \phi_{00}^{(2)} \cdot M^{1,2} \end{cases}, \qquad (35)$$

where $X^{0,1}$, $X^{1,2}$, and $M^{1,2}$ are the response amplitudes of the removed connections; $\alpha_{ij}^{(s)}$, $\beta_{ij}^{(s)}$, $\gamma_{ij}^{(s)}$, and $\phi_{ij}^{(s)}$ are the harmonic influence coefficients of subsystems $s$, namely $\alpha_{ij}^{(s)}$ and $\beta_{ij}^{(s)}$ as the displacement amplitudes and the rotation angle at the $i$-th point from the unit harmonic force applied at the $j$-th point; $\gamma_{ij}^{(s)}$ and $\phi_{ij}^{(s)}$ are the displacement amplitudes and the rotation angle at the $i$-th point from the unit harmonic moment applied at the $j$-th point; $\alpha_t^{(0)}$ is the harmonic influence coefficient of the tool subsystem represented as a system with one degree of freedom: $\alpha_t^{(0)} = \frac{1}{k_t^{(0)} - m_t^{(0)} \cdot \omega^2}$; $\omega$ is the vibration frequency [49].

After transforming Equations (34) and (35), the general system of deformation jointness equations at the points of subsystem disconnections will be of the form:

$$
\begin{cases}
(\alpha_{11}^{(1)} + \frac{1}{k_p^{(1)}} + \alpha_t^{(0)}) \cdot X^{0,1} + \alpha_{12}^{(1)} \cdot X^{1,2} + \gamma_{12}^{(1)} \cdot M^{1,2} = -\alpha_{11}^{(1)} \cdot F_1^{(1)} \\
\alpha_{21}^{(1)} \cdot X^{0,1} + (\alpha_{22}^{(1)} + \frac{1}{k_r^{(1)}} + \alpha_{00}^{(2)}) \cdot X^{1,2} + (\gamma_{22}^{(1)} + \gamma_{00}^{(2)}) \cdot M^{1,2} = -\alpha_{21}^{(1)} \cdot F_1^{(1)} \\
\beta_{21}^{(1)} \cdot X^{0,1} + (\beta_{22}^{(1)} + \beta_{00}^{(2)}) \cdot X^{1,2} + (\phi_{22}^{(1)} + \frac{1}{k_a^{(1)}} + \phi_{00}^{(2)}) \cdot M^{1,2} = -\beta_{21}^{(1)} \cdot F_1^{(1)}
\end{cases}
\tag{36}
$$

or

$$
[\mathbf{D}(\omega)] \cdot [\mathbf{X}] = [\Delta_F],
\tag{37}
$$

where $[\mathbf{X}]$ is the vector of harmonic response amplitudes of the removed connections $[\mathbf{X}] = (X^{0,1}, X^{1,2}, M^{1,2})^T$; $[\Delta]$ is the vector of displacement amplitudes from the action of the external load: $[\Delta] = \left(-\alpha_{11}^{(1)} \cdot F_1^{(1)}, -\alpha_{21}^{(1)} \cdot F_1^{(1)}, -\beta_{21}^{(1)} \cdot F_1^{(1)}\right)^T$; $[\mathbf{D}(\omega)]$ is the matrix of dynamic compliance (matrix of harmonic influence coefficients; see Equation (18)):

$$
[\mathbf{D}(\omega)] = \begin{pmatrix}
(\alpha_{11}^{(1)} + \frac{1}{k_p^{(1)}} + \alpha_t^{(0)}) & \alpha_{12}^{(1)} & \gamma_{12}^{(1)} \\
\alpha_{21}^{(1)} & (\alpha_{22}^{(1)} + \frac{1}{k_r^{(1)}} + \alpha_{00}^{(2)}) & (\gamma_{22}^{(1)} + \gamma_{00}^{(2)}) \\
\beta_{21}^{(1)} & (\beta_{22}^{(1)} + \beta_{00}^{(2)}) & (\phi_{22}^{(1)} + \frac{1}{k_a^{(1)}} + \phi_{00}^{(2)})
\end{pmatrix},
\tag{38}
$$

The main diagonal elements of the matrix $[\mathrm{D}(\omega)]$ correspond to the receptivity coefficients of the RCSA method at the point of separation of the subsystems [29]. From the equations system (36), the reactions of the removed connections are determined. Then, the transverse displacement amplitudes ($i = 0$–$u$) of the subsystems' characteristic points are determined:

- workpiece subsystem ($s = 1$):

$$
q_i^{(1)} = (\alpha_{i1}^{(1)} + F_1^{(1)}) \cdot X^{0,1} + \alpha_{i2}^{(1)} \cdot X^{1,2} + \gamma_{i2}^{(1)} \cdot M^{1,2},
\tag{39}
$$

- spindle chuck subsystem ($s = 2$):

$$
q_i^{(2)} = -\alpha_{i0}^{(2)} \cdot X^{0,1} - \gamma_{i0}^{(2)} \cdot M^{1,2},
\tag{40}
$$

- tool subsystem ($s = 0$):

$$
q_t^{(0)} = -\alpha_t^{(0)} \cdot X^{0,1},
\tag{41}
$$

The workpiece and tool subsystems are directly loaded with the force $F_1^{(1)}$ for an open dynamic model. The additional elastic coupling with stiffness $k_p$ is not taken into account in this case. In this case, the workpiece and tool subsystems are considered separately. Only the workpiece subsystem is decomposed. It consists of two subsystems—1st and 2nd.

Therefore, the condition of joint deformation is entered only for the 1st and 2nd subsystems, and dependencies (36) and (38) are represented in the forms:

$$
\begin{cases}
(\alpha_{22}^{(1)} + \frac{1}{k_r^{(1)}} + \alpha_{00}^{(2)}) \cdot X^{1,2} + (\gamma_{22}^{(1)} + \gamma_{00}^{(2)}) \cdot M^{1,2} = -\alpha_{21}^{(1)} \cdot F_1^{(1)} \\
(\beta_{22}^{(1)} + \beta_{00}^{(2)}) \cdot X^{1,2} + (\phi_{22}^{(1)} + \frac{1}{k_a^{(1)}} + \phi_{00}^{(2)}) \cdot M^{1,2} = -\beta_{21}^{(1)} \cdot F_1^{(1)}
\end{cases} \quad , \tag{42}
$$

$$
[\mathbf{D}(\omega)] = \begin{pmatrix}
(\alpha_{22}^{(1)} + \frac{1}{k_r^{(1)}} + \alpha_{00}^{(2)}) & (\gamma_{22}^{(1)} + \gamma_{00}^{(2)}) \\
(\beta_{22}^{(1)} + \beta_{00}^{(2)}) & (\phi_{22}^{(1)} + \frac{1}{k_a^{(1)}} + \phi_{00}^{(2)})
\end{pmatrix}, \tag{43}
$$

The natural frequencies of the spindle–workpiece system are determined from the condition $det\,[\mathbf{D}(\omega)] = 0$ using expression (36) for the closed model or expression (43) for the open-loop model. The natural frequency of the tool subsystem is defined by the dependence $f_t = \sqrt{\frac{k_t^{(0)}}{m_t^{(0)}}}$. The procedure for determining the harmonic influence coefficients $\alpha_{ij}^{(s)}$, $\beta_{ij}^{(s)}$, $\gamma_{ij}^{(s)}$, and $\phi_{ij}^{(s)}$ included in the matrices $[\mathbf{D}(\omega)]$ (38) and (43) is presented in Appendix B.

### 3.3.2. Experimental Determination of Natural Vibration Frequencies of the "Spindle–Workpiece–Tool" Elastic System

To measure the natural frequencies vibration, it is necessary to set the pressure force (thrust force $F_T$) on the workpiece with the tool (see Section 2.1 and Figure 1). This force is determined using direct experimental measurements of cutting forces during orthogonal cutting, as well as using the analytical cutting model (see Section 2.2.2) and numerical simulation of the orthogonal cutting process (see Section 2.2.3). Figure 10 shows a comparison of cutting forces obtained by these methods. The cutting forces are recalculated to the cutting width $w = 1$ mm. The measured value of the thrust force $F_T$ was used in experimental studies to determine the natural vibration frequencies of the elastic system "spindle–workpiece–tool". This force value was set when the tool was pressed to the workpiece and this force was monitored by the dynamometer signal (see Section 2.1 and Figure 1).

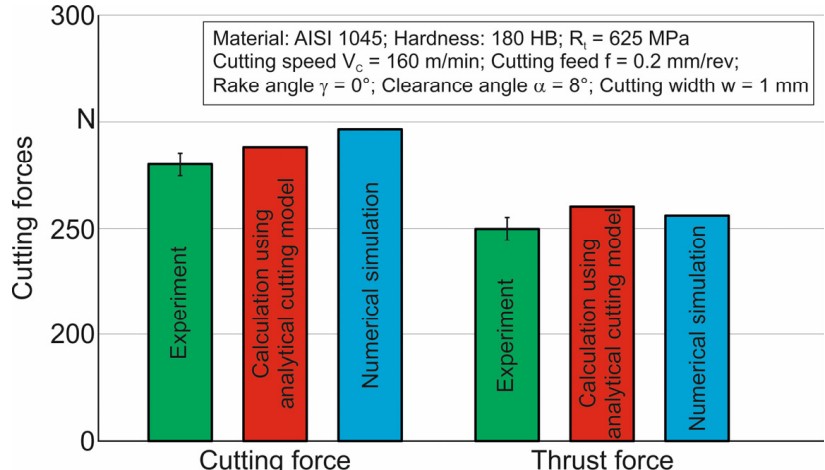

**Figure 10.** Comparative values of cutting forces and thrust forces determined by measurement using an analytical cutting model and via numerical simulation.

The influence of the contact force interaction between the workpiece and the tool on the natural vibration frequencies of the elastic "spindle–workpiece–tool" system was studied using simulation and verified experimentally. The natural frequencies spectra

of solid and tubular workpieces without tool pressure applied to the workpieces were obtained by measurements and the subsequent transformation of the measurement results. These spectra of the workpiece vibrations are shown in Figure 11.

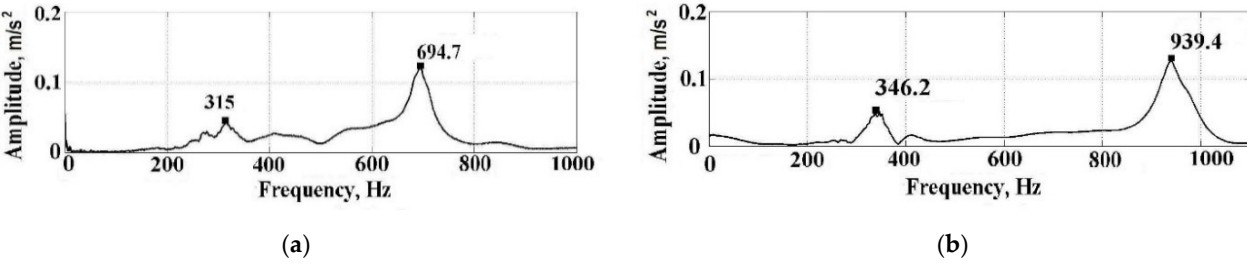

(**a**)                                                         (**b**)

**Figure 11.** Workpiece natural vibration spectra without tool pressure: (**a**)—solid workpiece; (**b**)—tubular workpiece.

Similar spectra of vibrations were received for all five variants of workpiece clamping by the tool at distances $L_1$ = 122 mm, $L_2$ = 102 mm, $L_3$ = 82 mm, $L_4$ = 62 mm, and $L_5$ = 42 mm from the place of workpiece clamping in the three-jaw chuck. The corresponding spectra of the workpiece vibrations were also obtained by calculating (19) the workpiece displacement amplitudes at the point of application of an external harmonic force. As an example, Figure 12 shows the experimental vibration spectra at overhang $L_2$ = 102 mm for tubular and solid workpieces.

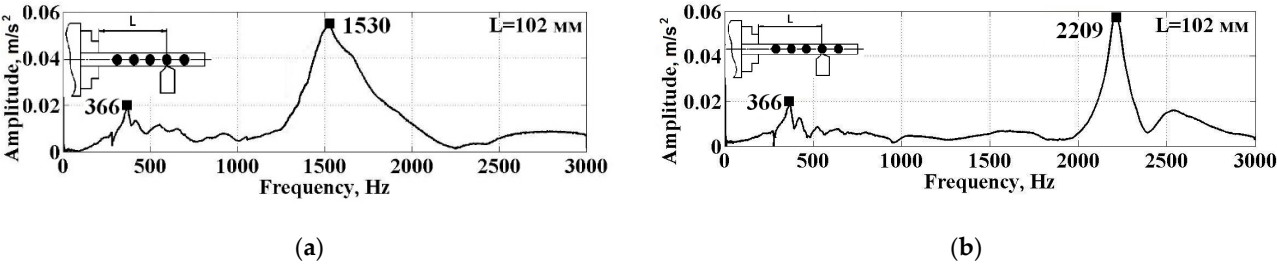

(**a**)                                                         (**b**)

**Figure 12.** Workpiece natural frequencies spectra with the tool pressure on the overhang $L_2$ = 102 mm: (**a**)—solid workpiece; (**b**)—tubular workpiece.

Histograms with comparative values of experimental and calculated natural frequencies for all studied overhangs are presented in Figure 13.

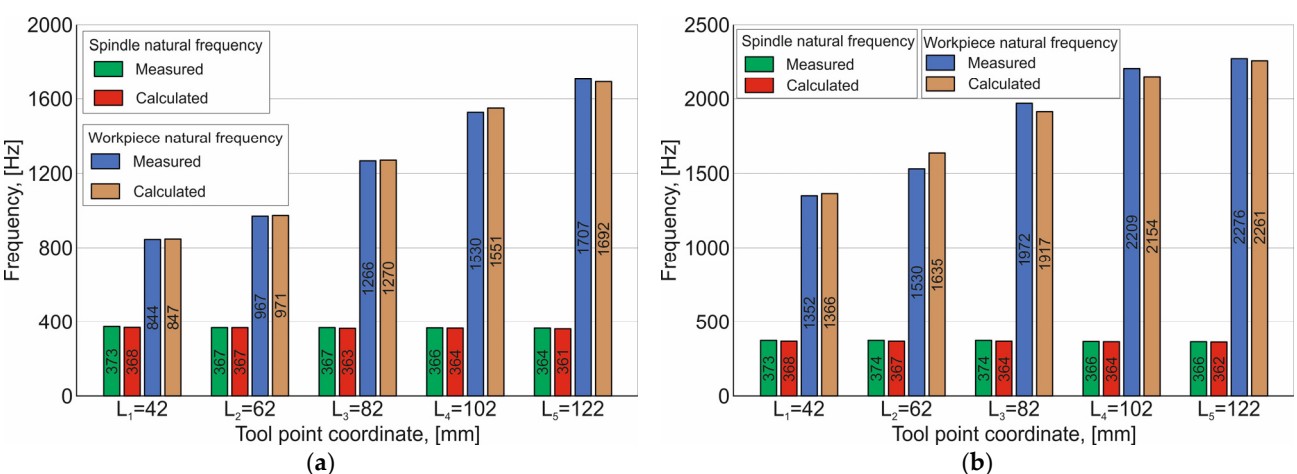

(**a**)                                                         (**b**)

**Figure 13.** Histograms of the workpiece's natural frequencies when the tool is pressed at a distance $L$ from the chuck: (**a**)—solid workpiece; (**b**)—tubular workpiece.

The natural frequency spectra of the workpiece without tool pressure (see Figure 11) and with tool pressure (see Figure 12) can be clearly defined as the two natural frequency zones of the "spindle–workpiece" system: the spindle (in the range of 315 Hz to 366 Hz) and the workpiece (above 690 Hz). The natural frequencies of the cantilevered fixed workpiece (see Figure 11) depend on their mass–inertial characteristics and are 695 Hz for a solid workpiece and 940 Hz for a tubular workpiece. When the workpiece is pressed with the tool at the coordinate point $L_2 = 102$ mm (see Figure 11), the frequency of the workpiece natural vibrations increases significantly, up to 1530 Hz for a solid workpiece and 2209 Hz for a tubular workpiece. The natural frequencies of the spindle increase insignificantly, from 315 Hz to 366 Hz for a solid workpiece and from 346 Hz to 366 Hz for a tubular workpiece. The histograms of the natural frequencies when the workpiece is pressed with a tool at the point coordinate $L$ (see Figure 13) confirm this trend for all tool point coordinates: $L_1 = 122$ mm, $L_2 = 102$ mm, $L_3 = 82$ mm, $L_4 = 62$ mm, and $L_5 = 42$ mm. A complete overview of the formation patterns and changes in the natural frequencies of the spindle and the workpiece, depending on the tool point coordinate for the two types of workpieces, is given by the cascade diagrams of the natural frequency spectrum for the workpiece when the tool point coordinate equals $L_i$. The cascade diagrams are shown in Figure 14.

The natural vibration frequencies of the "spindle–workpiece–tool" system were calculated according to the results from modeling the dynamic compliance matrix $[\mathbf{D}(\omega)]$ ((38) or (43)) from the condition $[\mathbf{D}(\omega)] = 0$. For a closed system (taking into account the tool pressure on the workpiece), $[\mathbf{D}(\omega)]$ was determined using (36). For an open-loop "spindle–workpiece" system (without tool pressure on the workpiece), $[\mathbf{D}(\omega)]$ was determined using (43). The numerical values of the coefficients assumed in the calculations are presented in Table 2.

The stiffness $k_p^{(1)}$ of the additional elastic connection was $= 0.404$ N/μm. This stiffness was calculated from the pressure force $F_T$ applied to the workpiece by the tool and the depth of the cut $a_{pr}$ (see (26)).

**Table 2.** Numerical values of the dynamic model coefficients (Figure 8).

| Dynamic Model Coefficients | Bearings | | | Tool–Holder–Chuck Joint |
|---|---|---|---|---|
| | Double-Row Cylindrical Roller Bearing (Position 1) | Angular Contact Ball Bearing (Positions 2 and 3) | Radial Bearing (Position 4) | |
| Radial stiffness, $k_i^r$ (N/μm) | 502 | 257 | 365 | 12 |
| Angular stiffness, $k_i^a$ (N·μm/rad) | 12,050 | - | - | $3.67 \times 10^{-2}$ |
| Damping, $h_i$ (N·s/mm) | 2 | 2 | 2 | 0.3 |
| lathe carriage | | | | |
| Equivalent mass, $m_c$ (kg) | 0.95 | | | |
| Equivalent stiffness, $k_c$ (N/μm) | 242 | | | |

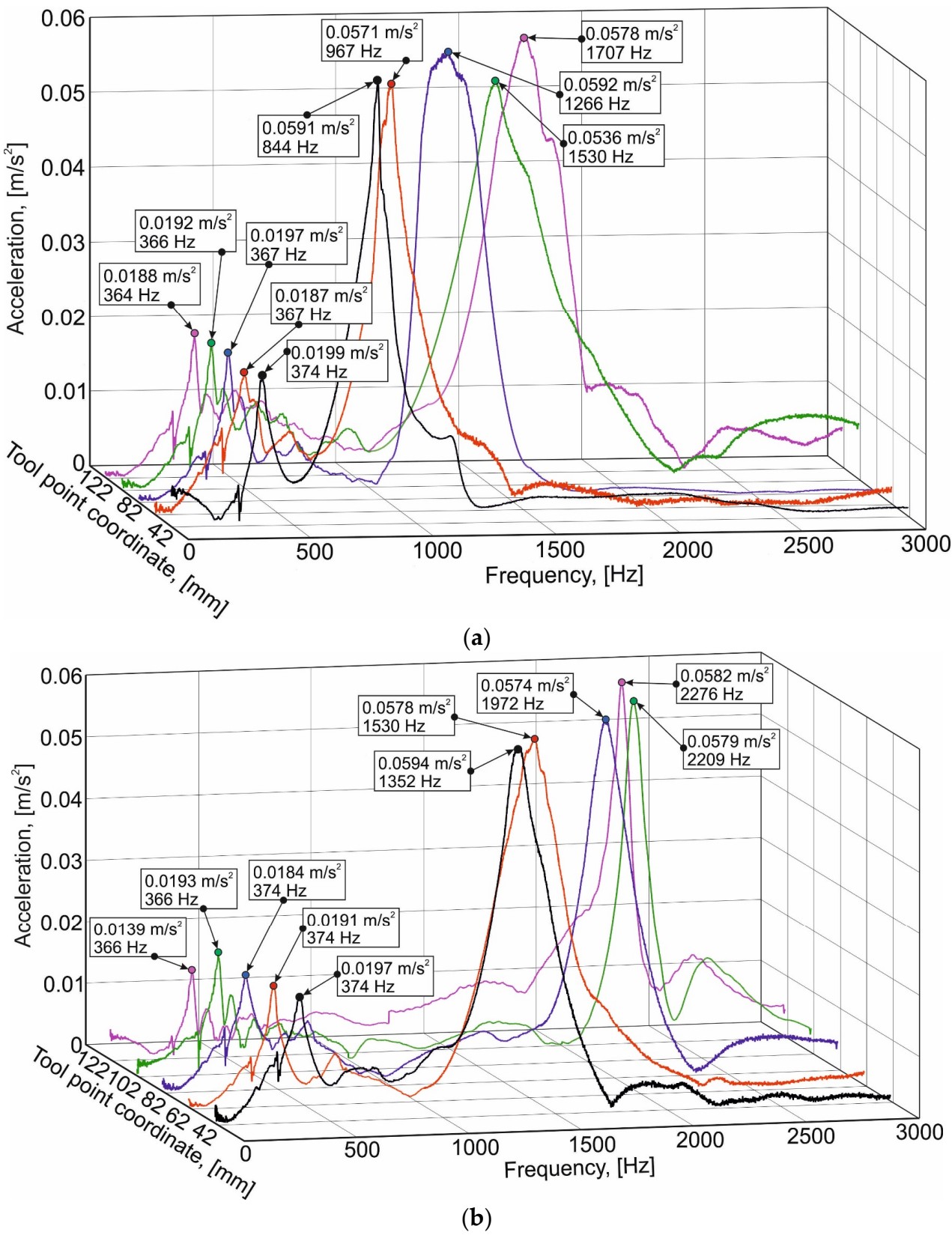

**Figure 14.** Cascade diagrams for the natural frequency spectra of the workpiece at different tool point coordinates $L_i$: (**a**)—solid workpiece; (**b**)—tubular workpiece.

## 4. Discussion

Using experimental measurements, the presence of two zones of vibrations with increased amplitudes in the frequency range from 0 to 1000 Hz was established (see Figure 11). These vibrations correspond to the first two natural frequencies of the system. In this case, the values of the natural frequencies are different for tubular and solid workpieces. For a solid workpiece, the first natural frequency is 346 Hz, and for a tubular workpiece, it is 315 Hz. The second natural frequency of the system changes significantly and is 694.7 Hz for a solid workpiece and 939.4 Hz for a tubular workpiece. At the same time, the natural frequencies did not change either when changing the sensor location on the workpiece or when changing the location of the impulse excitation application. The first natural frequency of the system changes insignificantly when the tool presses the workpiece: in the range of 366–374 Hz for a tubular workpiece and 364–373 Hz for a solid workpiece (see Figure 13). As the tool point coordinate decreases, the value of the first natural frequency increases. In addition, the values of the first natural frequency of the system (366 Hz) when the workpiece is pressed with the tool (see Figure 12) are somewhat higher than the corresponding values of the first natural frequency (315 and 344.2 Hz) of the nonclamped workpiece (see Figure 11). The second natural frequency of the system (see Figure 13) changes much more significantly. The frequency value decreases as the tool point coordinate decreases: from 2276 Hz to 1352 Hz for a tubular workpiece and from 1707 Hz to 844 Hz for a solid workpiece.

The first and second natural frequencies of the spindle with chuck subsystem 2 are 349.5 Hz and 1563 Hz, respectively. The first natural frequency of the workpiece subsystem 1, taking into account the rigidity of its fixing in the chuck, is equal to a tubular workpiece with 949 Hz and for a solid workpiece with 743 Hz (see Figure 11). The first natural frequency calculated values of the "spindle–workpiece" system for the tubular and solid workpieces are 331 Hz and 318 Hz, respectively. These frequencies are close to the calculated value of the first natural frequency of the "spindle–chuck" subsystem 2 (349.5 Hz). Thus, the system's first natural frequency (see Figure 11) is the first natural frequency of the "spindle–chuck" subsystem 2. The calculated values of the second natural frequency of the "spindle–workpiece" system for the tubular and solid workpieces are 937 Hz and 721 Hz, respectively. These frequencies are close to the calculated values of the first natural frequency of workpiece subsystem 1 (949 Hz and 743 Hz, respectively). Thus, the system's second natural frequency is the first natural frequency of workpiece subsystem 1. The insignificant change in the system's first natural frequency when clamping different workpieces in the chuck without taking into account the tool pressure (331 Hz for a tubular workpiece and 318 Hz for a solid workpiece; see Figure 11) is explained by a minor contribution of the workpiece in changing the mass–inertial characteristics of the "spindle–workpiece" system. The significant change in the system's second natural frequency under the same conditions (937 Hz for a tubular workpiece and 721 Hz for a solid workpiece) is explained by a significant (as compared to the change in the natural stiffness) change in the mass–inertial characteristics of the workpiece itself.

The spindle's first natural frequency with the workpiece clamped in the chuck and the tool pressed is in the range of 361–368 Hz (see Figure 13). This frequency increases insignificantly as the tool point coordinate decreases and does not depend on the workpiece type. This confirms that the change in the workpiece type has little effect on the change in the first natural frequency of the "spindle–workpiece" system. The second natural frequency of the spindle with the workpiece clamped in the chuck and the tool pressed depends significantly on both the workpiece type and the tool point coordinate (see Figure 13). The second natural frequency decreases as the tool point coordinate decreases: from 2261 to 1366 Hz for a tubular workpiece and from 1692 to 847 Hz for a solid workpiece. This is explained by the fact that the tool in a closed dynamic system acts as an additional support, and its location significantly affects the workpiece natural frequencies. Thus, when the tool point coordinate decreases, the length of the cantilevered workpiece part increases, and accordingly, the value of the second natural frequency decreases. The calculation results of

both the character and values correspond to the experimental results (see Figure 13). The calculation error of the natural frequencies for the workpiece clamped in the chuck without taking into account its tool pressure does not exceed 1.4% and, when taking into account the tool pressure, does not exceed 6.5%. It should also be noted that the main trends in the dynamic behavior of the cantilevered workpiece under conditions simulating length turning coincide with the results of similar studies of the length turning dynamics with a workpiece clamped in centers [33,34].

As a result of this study, the low dynamic sensitivity of the spindle assembly to changes in the type and loading character of the workpiece was established. On the one hand, this confirms the possibility of replacing the analytical determination results of the structural dynamic responses of the forming units (workpiece and tool) with experimental ones for practical applications [34]. On the other hand, changes in the spindle dynamic characteristics, even within a small range, indicate that the dynamic interaction between the spindle and the workpiece (tool) has not been sufficiently studied. This may, for example, concern the quantification of the stiffness and damping in the workpiece or tool clamping unit or the influence of individual spindle bearing parameters on the FRF determination [34].

## 5. Conclusions

The basic condition for the decomposition of the multicomponent closed dynamic system of the machine tool is determined by the type and character of the relation between the partial subsystems of the workpiece and the tool in the cutting zone. The presentation of the cutting process in the form of an elastic relation between the workpiece and the tool ensures that this basic condition is fulfilled.

The developed dynamic model of the "spindle–tool–tool system" ensures a combined account of the coordinate coupling of the conjugated partial subsystems and the dynamic character of the elastic coupling between them. This account is implemented by means of the method of dynamic compliance to decompose the system and the matrix transition method to determine the coefficients of the subsystems' harmonic influence.

The coordinate coupling in the developed model is implemented by the coincidence equations of deformations at the opening points of the coupled partial subsystems. The dynamic character of the elastic relation between these subsystems is defined by the coupled vibrations of the contacting bodies. The relation of these vibrations is ensured by determining the harmonic influence coefficients of the subsystems. This approach is applicable to all coupled partial subsystems of the machine tool mechanical system, including the "workpiece–tool" pair.

The adequacy of the developed dynamic model of the system "spindle–workpiece–tool" is evaluated based on the measurement results and modeling of the natural vibration frequencies of the dynamic system. The adequacy test was performed for two types of workpieces cantilevered in the spindle chuck at fixed length locations of their loading with the cutting tool. This ensured that the main factors determining the formation and change in the dynamic characteristics of the machine's mechanical system were taken into account.

Through measurements and calculations of the system "spindle–workpiece–tool" vibrations, it was found that the natural vibration frequency of the workpiece significantly depends on its type, as well as the condition and point coordinates of its pressure point with the tool. The spindle natural frequency under the same conditions changes insignificantly. The deviation between the measured and calculated natural frequency values of the workpiece clamped in the chuck without taking into account its pressure with the tool does not exceed 1.4%. Taking into account the pressure of the workpiece with the tool, this deviation does not exceed 6.5%.

Experimental verification of the calculation results of the system's natural frequencies proved the correctness of the tendency to change these frequencies depending on the changes in the mass–inertial characteristics of the workpiece and the conditions of its interaction with the cutter. This confirms the advisability of representing the contact force

interaction between the tool and the workpiece in the calculation models as an additional movable elastic support.

**Author Contributions:** Conceptualization, Y.D. and M.D.; methodology, Y.D.; software, A.P.; validation, Y.D. and A.P.; formal analysis, M.S.; investigation, M.D.; resources, Y.D.; data curation, A.P.; writing—original draft preparation, Y.D. and M.S.; writing—review and editing, Y.D. and M.S.; visualization, A.P.; and project administration, Y.D. and M.S. All authors have read and agreed to the published version of the manuscript.

**Funding:** This research received no external funding.

**Data Availability Statement:** Not applicable.

**Acknowledgments:** The authors would like to thank the Department of Mechanical Engineering Technology Igor Sikorsky Kyiv Polytechnic Institute for its support in the preparations of the experimental studies, which is highly appreciated.

**Conflicts of Interest:** The authors declare no conflict of interest.

## Nomenclature

| | | | |
|---|---|---|---|
| $W(\omega)$ | dynamic transfer function | $\theta_i$ | the slope of the subsystem in *i*-th section |
| $\widetilde{F}_1^{(1)}$ | cutting force in the form of polyharmonic functions with zero harmonic component | $M$ | external bending moment in *i*-th section |
| $F_s^{(1)}$ | the static component of the $\widetilde{F}_1^{(1)}$ | $Q$ | external shear force in *i*-th section |
| $F_d^{(1)}$ | the dynamic component of the $\widetilde{F}_1^{(1)}$ | $\{\mathbf{Y}\}_i$ | subsystem parameters column vector |
| $a$ | chip width | $[\mathbf{T}]$ | general transfer matrix of a subsystem, which is equal to the product of section transfer matrices $[\mathbf{T}]_i$ by order from beam end to its beginning |
| $K_f$ | cutting coefficient | $[\mathbf{T}]_i$ | transfer matrix of subsystem section $(4*4)$ between *i*-1 and *i* points |
| $k_p^{(1)}$ | cutting process stiffness | $[\mathbf{T}_p]_i$ | mass and inertia matrix of localized mass |
| $k_{r,i}^{(s)}$ | radial stiffness of subsystems linkage | $[\mathbf{T}_b]_i$ | elastic support matrix with damping |
| $k_{a,i}^{(s)}$ | angular stiffness of subsystems linkage | $[\mathbf{T}_u]_i$ | beam section matrix with distributed mass |
| $k_t^{(1)}$ | elastic linkage stiffness of the workpiece $s=1$ and carriage $s=0$ subsystems | $[\mathbf{T}_i]_j$ | matrix, which is equal to the product of section transfer matrices $[\mathbf{T}_i]$, between *i*-th and *j*-th sections |
| $X_i^{s,s+1}$, $M_i^{s,s+1}$ | reactions of discarded bonds in the *i*-th point of subsystems *s* and *s* + 1 disconnection, forces, and moments | $[\mathbf{T}_i]_0$ | matrix, which is equal to the product of section transfer matrices $[\mathbf{T}_i]$, located between *i*-th and 0-th sections |
| $\alpha_{ij}^{(s)}$, $\beta_{ij}^{(s)}$ | displacement-to-force and slope-to-force receptances in *i*-th section from the unit harmonic force in *j*-th section | $P, S, R, T$ | functions |
| $\gamma_{ij}^{(s)}$, $\phi_{ij}^{(s)}$ | displacement-to-moment and slope-to-moment receptances in *i*-th section from the unit harmonic moment in *j*-th section | $m_w$ | mass of the weight |
| $\alpha_{uu}^{(1,2)}$, $\phi_{uu}^{(1,2)}$ | displacement-to-force and slope-to-moment receptances at separation points of $s=1$ and $s=2$ subsystems | $J$ | transverse moment of inertia of localized mass |

| | | | |
|---|---|---|---|
| $\alpha_t^{(0)}$ | displacement-to-force receptance of the tool subsystem, represented as a 1-DOF (degree of freedom) system | $[\mathbf{F}]_j$ | load column vector in *j*-th section $[\mathbf{F}]_j = \{0,0,0,P_j\}^T$ or $[\mathbf{F}]_j = \{0,0,M_j,0\}$ |
| $[\mathbf{F}]$ | block/partitioned matrix of discarded bonds generalized reaction amplitudes **X** (forces) and **M** (bending moments) from external harmonic loads | $a_{ab}$ | [**T**] matrix elements |
| $[\mathbf{D}(\omega)]$ | receptance block/partitioned matrix | $H_s$ | specified cutting depth |
| $[\mathbf{\Delta_F}]$ | block/ partitioned matrix of generalized receptances from external harmonic load | $H_a$ | actual cutting depth |
| **X** | column vectors of discarded bonds, forces | $C_p, k$ | correction factors |
| **M** | bending moments amplitudes | $S$ | feed |
| $q_i^{(s)}$ | calculated transverse (radial) displacements function of subsystem *s* in the *i*-th point | | |
| $y$ | transverse displacement of the beam with x coordinate | $V$ | cutting speed |
| $x$ | coordinate in the direction of the longitudinal axis of the beam | $x, y, n$ | indices |
| $E\,I_i$ | the flexural stiffness of the beam section | $\widetilde{q}_1^{(1)}, \widetilde{q}_c^{(0)}$ | workpiece and tool elastic displacements in the cutting zone |
| $m_i$ | weight of the unit length of the beam section | $\widetilde{\alpha}_c^{(0)}, \widetilde{\alpha}_{i1}^{(1)}, \widetilde{\alpha}_{i2}^{(1)}, \widetilde{\gamma}_{i2}^{(1)}$ $\widetilde{X}^{0,1}, \widetilde{X}^{1,2}\, \widetilde{M}^{1,2}$ | generalized subsystems receptance reactions of discarded bonds in the form of polyharmonic functions with zero harmonic; components $\left( \widetilde{X}^{0,1} = X_m^{0,1} + X_v^{0,1} \right.$, $\widetilde{M}^{1,2} = M_m^{0,1} + M_v^{0,1} )$ |
| $t$ | time | $f_t$ | tool natural frequency |
| $l$ | beam section length | $f_s^{(12)}, f_s^{(12)}$ | first and second natural frequencies of the closed-loop dynamic system |
| $\omega$ | the angular frequency of the bending vibration of the beam | $s$ | subsystem index, *s* = 0,1,2,3 |
| $F_j$ | external harmonic force in *j*-th section | $u$ | number of subsystem sections |
| $M_j$ | an external harmonic moment in *j*-th section | $h_i^{(s)}$ | subsystem s damping coefficient in *i*-th cross-section |
| $y_i$ | deflection of the subsystem in *i*-th section | $m_c^{(0)}$ | localized mass of the tool subsystem *s* = 0 |

## Appendix A

Depending on the place of external load application (location of the *j*-th section relative to the calculated *u*-th section), the $[\mathbf{T}]_{u,j}$ matrix is determined by the equations:

$$[\mathbf{T}]_{u,j} = \begin{cases} \prod_{i=u}^{j+1} [\mathbf{T}]_i & for\ 0 < j < u \\ [\mathbf{T}] = \prod_{i=u}^{0} [\mathbf{T}]_i & for\ j = 0 \\ diag(1,1,1,1) & for\ j = u \end{cases} ; \tag{A1}$$

Thus, for the beam shown in Figure [7], $[\mathbf{T}]_{u,j} = [\mathbf{T}]_u \cdot [\mathbf{T}]_k = [\mathbf{T}_u]_u \cdot [\mathbf{T}_b]_k \cdot [\mathbf{T}_u]_k$. For the case of beam loading by harmonic force (see Figure [7]a), the matrix Equation (24) is expanded as follows:

$$
\begin{bmatrix} y_u \\ \theta_u \\ 0 \\ 0 \end{bmatrix} = \begin{bmatrix} a_{11} & a_{12} & a_{13} & a_{14} \\ a_{21} & a_{22} & a_{23} & a_{24} \\ a_{31} & a_{32} & a_{33} & a_{34} \\ a_{41} & a_{42} & a_{43} & a_{44} \end{bmatrix} \cdot \begin{bmatrix} y_0 \\ \theta_0 \\ 0 \\ 0 \end{bmatrix} + \begin{bmatrix} a_{11}^{u,j} & a_{12}^{u,j} & a_{13}^{u,j} & a_{14}^{u,j} \\ a_{21}^{u,j} & a_{22}^{u,j} & a_{23}^{u,j} & a_{24}^{u,j} \\ a_{31}^{u,j} & a_{32}^{u,j} & a_{33}^{u,j} & a_{34}^{u,j} \\ a_{41}^{u,j} & a_{42}^{u,j} & a_{43}^{u,j} & a_{44}^{u,j} \end{bmatrix} \cdot \begin{bmatrix} 0 \\ 0 \\ 0 \\ F_j \end{bmatrix} \tag{A2}
$$

where $a_{ab}$ and $a_{ab}^{u,j}$ are the elements of matrices $[\mathbf{T}]$ and $[\mathbf{T}]_{u,j}$ defined by dependencies (7) and (25), respectively.

From the last two equations of the system (A2), the parameters $y_0$ and $\theta_0$ in the 0-th beam section and the corresponding influence coefficients $\alpha_{0j}$ and $\beta_{0j}$ are determined:

$$
\begin{cases} \alpha_{0j} = \dfrac{y_0}{F_j} = \dfrac{a_{32} \cdot a_{44}^{u,j} - a_{42} \cdot a_{34}^{u,j}}{a_{31} \cdot a_{42} - a_{41} \cdot a_{32}} ; \\ \beta_{0j} = \dfrac{\theta_0}{F_j} = \dfrac{a_{41} \cdot a_{34}^{u,j} - a_{31} \cdot a_{44}^{u,j}}{a_{31} \cdot a_{42} - a_{41} \cdot a_{32}} \end{cases} \tag{A3}
$$

The influence coefficients $y_{0j}$ and $\varphi_{0j}$ when loading the beam with a harmonic moment (see Figure [7]b) are determined similarly:

$$
\begin{cases} \gamma_{0j} = \dfrac{y_0}{M_j} = \dfrac{a_{32} \cdot a_{43}^{u,j} - a_{42} \cdot a_{33}^{u,j}}{a_{31} \cdot a_{42} - a_{41} \cdot a_{32}} ; \\ \phi_{0j} = \dfrac{\theta_0}{M_j} = \dfrac{a_{41} \cdot a_{33}^{u,j} - a_{31} \cdot a_{43}^{u,j}}{a_{31} \cdot a_{42} - a_{41} \cdot a_{32}} \end{cases} \tag{A4}
$$

The influence coefficients in the other sections of the rod are determined using a matrix equation connecting the parameters of the arbitrary *i*-th section with the 0-th section:

$$
\{\mathbf{Y}\}_i = [\mathbf{T}]_{i,0} \cdot \{\mathbf{Y}\}_0 + [\mathbf{T}]_{i,j} \cdot \{\mathbf{F}\}_j; \tag{A5}
$$

where $\{\mathbf{Y}\}_i = \{y_i, \theta_i, M_i, Q_i\}^T$ is the vector of amplitudes of transverse displacement $y_i$, rotation angle $\theta_i$, bending moment $M_i$, and transverse force $Q_i$ in the *i*-th section of the beam; $[\mathbf{T}]_{i,0}$ is a matrix equal to the product of the transfer matrices $[\mathbf{T}]_i$ of the sections placed between the *i*-th and 0-th sections; $[\mathbf{T}]_{i,j}$ is a matrix equal to the product of the transfer matrices $[\mathbf{T}]_i$ of the sections placed between the *i*-th and *j*-th sections.

The $[\mathbf{T}]_{i,0}$ and $[\mathbf{T}]_{i,j}$ matrices are determined by the dependencies:

$$
[\mathbf{T}]_{i,0} = \prod_{i=i}^{0} [\mathbf{T}]_i; \quad [\mathbf{T}]_{i,j} = \begin{cases} \prod_{i=i}^{j+1} [\mathbf{T}]_i & for \ 0 < j < i \\ [\mathbf{T}]_{i,0} = \prod_{i=i}^{0} [\mathbf{T}]_i & for \ j = 0 \\ diag(1,1,1,1) & for \ j = i \\ 0 & for \ j > i \end{cases} ; \tag{A6}
$$

Thus, for the beam shown in Figure [7]:

- for the *k*-th section:

$$
[\mathbf{T}]_{k,j} = [\mathbf{T}]_k \ \text{и} \ [\mathbf{T}]_{k,0} = [\mathbf{T}]_k \cdot [\mathbf{T}]_j \cdot [\mathbf{T}]_i \cdot [\mathbf{T}]_1;
$$

- for the *j*-th section:

$$
[\mathbf{T}]_{j,j} = diag(1,1,1,1) \ \text{и} \ [\mathbf{T}]_{j,0} = [\mathbf{T}]_j \cdot [\mathbf{T}]_i \cdot [\mathbf{T}]_1;
$$

- for the *i*-th section:

$$[\mathbf{T}]_{i,j} = 0 \text{ и } [\mathbf{T}]_{i,0} = [\mathbf{T}]_i \cdot [\mathbf{T}]_1;$$

- for the 1-st section:

$$[\mathbf{T}]_{k,j} = 0 \text{ и } [\mathbf{T}]_{1,0} = [\mathbf{T}]_1.$$

In the case of a beam loading with harmonic force (see Figure 7a), the matrix Equation (A5) in the expanded form is represented by the following equation:

$$
\begin{bmatrix} y_i \\ \theta_i \\ M_i \\ Q_i \end{bmatrix} =
\begin{bmatrix}
a_{11}^{i,0} & a_{12}^{i,0} & a_{13}^{i,0} & a_{14}^{i,0} \\
a_{21}^{i,0} & a_{22}^{i,0} & a_{23}^{i,0} & a_{24}^{i,0} \\
a_{31}^{i,0} & a_{32}^{i,0} & a_{33}^{i,0} & a_{34}^{i,0} \\
a_{41}^{i,0} & a_{42}^{i,0} & a_{43}^{i,0} & a_{44}^{i,0}
\end{bmatrix} \cdot
\begin{bmatrix} y_0 \\ \theta_0 \\ 0 \\ 0 \end{bmatrix} +
\begin{bmatrix}
a_{11}^{i,j} & a_{12}^{i,j} & a_{13}^{i,j} & a_{14}^{i,j} \\
a_{21}^{i,j} & a_{22}^{i,j} & a_{23}^{i,j} & a_{24}^{i,j} \\
a_{31}^{i,j} & a_{32}^{i,j} & a_{33}^{i,j} & a_{34}^{i,j} \\
a_{41}^{i,j} & a_{42}^{i,j} & a_{43}^{i,j} & a_{44}^{i,j}
\end{bmatrix} \cdot
\begin{bmatrix} 0 \\ 0 \\ 0 \\ F_j \end{bmatrix},
\tag{A7}
$$

where $a_{ab}^{i,0}$ and $a_{ab}^{i,j}$ are the elements of matrices $[\mathbf{T}]_{i,0}$ and $[\mathbf{T}]_{i,j}$ defined by dependencies (A6).

Using Equation (A7) and taking into account the systems of Equations (A3), the expressions for the influence coefficients $\alpha_{ij}$ and $\beta_{ij}$ are obtained:

$$
\begin{cases}
\alpha_{ij} = \frac{y_i}{F_j} = a_{11}^{i,0} \cdot \alpha_{0j} + a_{12}^{i,0} \cdot \beta_{0j} + a_{14}^{i,j} \\
\beta_{ij} = \frac{\theta_i}{F_j} = a_{21}^{i,0} \cdot \alpha_{0j} + a_{22}^{i,0} \cdot \beta_{0j} + a_{24}^{i,j}
\end{cases};
\tag{A8}
$$

The influence coefficients $y_{ij}$ and $\varphi_{ij}$ when loading the beam with a harmonic moment (see Figure 7b) are determined similarly:

$$
\begin{cases}
\gamma_{ij} = \frac{y_i}{M_j} = a_{11}^{i,0} \cdot \gamma_{0j} + a_{12}^{i,0} \cdot \phi_{0j} + a_{13}^{i,j} \\
\phi_{ij} = \frac{\theta_i}{M_j} = a_{21}^{i,0} \cdot \gamma_{0j} + a_{22}^{i,0} \cdot \phi_{0j} + a_{23}^{i,j}
\end{cases},
\tag{A9}
$$

**Appendix B**

The harmonic influence coefficients $\alpha_{ij}^{(s)}$, $\beta_{ij}^{(s)}$, $\gamma_{ij}^{(s)}$, and $\phi_{ij}^{(s)}$ of the subsystems (see Figure 9), included in the matrices $[\mathbf{D}(\omega)]$, are determined by dependencies (A3) and (A4) and (A8) and (A9) using supporting matrices (A1) and (A6). Determination of the harmonic influence coefficients $\alpha_{ij}^{(1)}$, $\beta_{ij}^{(1)}$, $\gamma_{ij}^{(1)}$, and $\phi_{ij}^{(1)}$ included in the matrix $[\mathbf{D}(\omega)]$ (36) the workpiece subsystem (*s* = 1, Figure 9) is performed as follows:

The workpiece subsystem (see Figure 9) consists of two sections (*u* = 2). Each of the sections contains only a beam element with distributed mass, which is described by the transfer matrix (5). Given the rules for the transfer matrices (7) and (8), the transfer matrix $[\mathbf{T}]^{(1)}$ of this subsystem equals:

$$
[\mathbf{T}]^{(1)} = \prod_{i=2}^{1} [\mathbf{T}]_i^{(1)} = [\mathbf{T}]_2^{(1)} \cdot [\mathbf{T}]_1^{(1)},
\tag{A10}
$$

where $[\mathbf{T}]_2^{(1)} = [\mathbf{T}_u]_2^{(1)}$ and $[\mathbf{T}]_1^{(1)} = [\mathbf{T}_u]_1^{(1)}$.

The supporting matrices $[\mathbf{T}]_{u,j}^{(1)}$, $[\mathbf{T}]_{i,0}^{(1)}$, and $[\mathbf{T}]_{i,j}^{(1)}$ of the workpiece subsystems are derived from the transfer matrix $[\mathbf{T}]^{(1)}$ (A10) and are calculated by the dependencies (A1) and (A6). The component elements of the supporting matrices are used in the dependencies (A3) and (A4) and (A8) and (A9) for direct calculation of the harmonic influence coefficients $\alpha_{ij}^{(1)}$, $\beta_{ij}^{(1)}$, $\gamma_{ij}^{(1)}$, and $\phi_{ij}^{(1)}$ of the workpiece subsystem. The calculation equations for the supporting matrices are given in Table A1.

**Table A1.** Supporting matrices for the workpiece subsystem ($s = 1$, $u = 2$).

| Harmonic Influence Coefficients of Type $\alpha_{ij}^{(1)}, \beta_{ij}^{(1)}, \gamma_{ij}^{(1)}, \phi_{ij}^{(1)}$ | Beam Section Number | | Supporting Matrices | | |
|---|---|---|---|---|---|
| | i | j | $[T]_{u,j}^{(1)}$ | $[T]_{i,0}^{(1)}$ | $[T]_{i,0}^{(1)}$ |
| 1 | 2 | 3 | 4 | 5 | 6 |
| $\alpha_{01}^{(1)}, \beta_{01}^{(1)}$ | 0 | 1 | $[\mathbf{T}]_3^{(1)}\cdot[\mathbf{T}]_2^{(1)}$ | - | - |
| $\alpha_{02}^{(1)}, \beta_{02}^{(1)}, \gamma_{02}^{(1)}, \phi_{02}^{(1)}$ | 0 | 2 | $[\mathbf{T}]_3^{(1)}$ | - | - |
| $\alpha_{11}^{(1)}$ | 1 | 1 | - | $[\mathbf{T}]_1^{(1)}$ | $diag(1,1,1,1)$ |
| $\alpha_{21}^{(1)}, \beta_{21}^{(1)}$ | 2 | 1 | - | $[\mathbf{T}]_2^{(1)}\cdot[\mathbf{T}]_1^{(1)}$ | $[\mathbf{T}]_2^{(1)}$ |
| $\alpha_{12}^{(1)}, \gamma_{12}^{(1)}$ | 1 | 2 | - | $[\mathbf{T}]_1^{(1)}$ | 0 |
| $\alpha_{22}^{(1)}, \beta_{22}^{(1)}, \gamma_{22}^{(1)}, \phi_{22}^{(1)}$ | 2 | 2 | - | $[\mathbf{T}]_2^{(1)}\cdot[\mathbf{T}]_1^{(1)}$ | $diag(1,1,1,1)$ |

The determination of the harmonic influence coefficients $\alpha_{00}^{(2)}$, $\beta_{00}^{(2)}$, $\gamma_{00}^{(2)}$, and $\phi_{00}^{(2)}$ of the spindle–chuck subsystem ($s = 2$, Figure 9) included in the matrix $[\mathbf{D}(\omega)]$ (36) is performed as follows. The spindle–chuck subsystem ($s = 2$) consists of eight sections ($u = 8$). The third, fourth, fifth, and seventh sections contain beam elements with distributed mass and elastic bearings. The remaining sections contain only beam elements with distributed mass. Given the rules for the transfer matrices (7) and (8), the transfer matrix $[\mathbf{T}]^{(2)}$ of this subsystem equals:

$$[\mathbf{T}]^{(2)} = \prod_{i=8}^{1}[\mathbf{T}]_i^{(2)} = [\mathbf{T}]_8^{(2)}\cdot[\mathbf{T}]_7^{(2)}\cdot[\mathbf{T}]_6^{(2)}\cdot[\mathbf{T}]_5^{(2)}\cdot[\mathbf{T}]_4^{(2)}\cdot[\mathbf{T}]_3^{(2)}\cdot[\mathbf{T}]_2^{(2)}\cdot[\mathbf{T}]_1^{(2)}, \tag{A11}$$

where $[\mathbf{T}]_8^{(2)} = [\mathbf{T}_u]_8^{(2)}$, $[\mathbf{T}]_7^{(2)} = [\mathbf{T}_b]_7^{(2)}\cdot[\mathbf{T}_u]_7^{(2)}$, $[\mathbf{T}]_6^{(2)} = [\mathbf{T}_u]_6^{(2)}$, $[\mathbf{T}]_5^{(2)} = [\mathbf{T}_b]_5^{(2)}\cdot[\mathbf{T}_u]_5^{(2)}$, $[\mathbf{T}]_4^{(2)} = [\mathbf{T}_b]_4^{(2)}\cdot[\mathbf{T}_u]_4^{(2)}$, $[\mathbf{T}]_3^{(2)} = [\mathbf{T}_b]_3^{(2)}\cdot[\mathbf{T}_u]_3^{(2)}$, $[\mathbf{T}]_2^{(2)} = [\mathbf{T}_u]_2^{(2)}$, $[\mathbf{T}]_1^{(2)} = [\mathbf{T}_u]_1^{(2)}$.

The harmonic influence coefficients $\alpha_{00}^{(2)}$, $\beta_{00}^{(2)}$, $\gamma_{00}^{(2)}$, and $\phi_{00}^{(2)}$ of the spindle–chuck subsystem are determined by dependencies (A3) and (A4) when $j = 0$. The natural frequencies of the spindle–chuck subsystem are determined from the condition of maximum dynamic compliance (the first equation of the system (A3)):

$$a_{31}\cdot a_{42} - a_{41}\cdot a_{32} = 0, \tag{A12}$$

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
