# Peer review of "Cutting Process Consideration in Dynamic Models of Machine Tool Spindle Units"

_machines, doi:10.3390/machines11060582_

Round 1

Reviewer 1 Report

Although the paper has successfully demonstrated that experimental results can replace the analysis of the dynamic response of a single unit structure for practical applications, there are many complex factors that affect actual machining, and the influence of vibration frequency also involves the entire lathe structure and other factors. Therefore, it is suggested that the author further verify its feasibility by comparing the vibration frequency of the entire machine tool with that of the "spindle-workpiece" system through experiments.

At the same time, the following points should be considered in the "spindle-workpiece" system:

1. Regarding the issue of cutting width, it is recommended that the author provide a more clear explanation and description in the paper. The author can provide more detailed experimental data and calculation processes to demonstrate the rationality of recalculating the width as 1mm in the cutting model and numerical simulation stage to avoid errors that may affect the conclusions of the paper.

2. Regarding the influence of spindle thermal deformation and tool wear on frequency, it is suggested that the author conduct more in-depth research and analysis on these factors, or explain them in the paper to fully consider the degree of influence of these factors, ensuring the accuracy and reliability of the conclusions.

3. Regarding the issue of the influence of vibration frequency in the measurement and calculation of "spindle-workpiece" vibration, it is recommended that the author provide more data and analysis in the paper to further demonstrate the validity and accuracy of the conclusions. For example, the author can compare experimental data under different conditions to demonstrate the influence of vibration frequency on pressure and tool position, and explore the influence of a single variable on the conclusions. In addition, it is suggested that the author clearly state the limitations and room for improvement of the conclusions to encourage further research and exploration.

Some English sentences and grammar can be improved in the total paper. 

Author Response

The authors are very grateful to Reviewer #1 for meticulously reviewing and interpreting the content of the paper.

Reviewer #1:

  1. Although the paper has successfully demonstrated that experimental results can replace the analysis of the dynamic response of a single unit structure for practical applications, there are many complex factors that affect actual machining, and the influence of vibration frequency also involves the entire lathe structure and other factors. Therefore, it is suggested that the author further verify its feasibility by comparing the vibration frequency of the entire machine tool with that of the "spindle-workpiece" system through experiments.
  • The developed model presented in the paper describes the dynamic interaction between the workpiece and tool in the cutting process and enables the theoretical determination of the numerical characteristics for interrelated vibrations of the conjugate elements of the machine tool structure. The model includes mechanical subsystems of the machine tool forming units that determine the main accuracy and quality parameters of the machining process. The model structure and method of its development ensure the possibility of expanding the number of mechanical subsystems, taking into account their influence on the relative vibrations of the tool and workpiece. Experimental studies of natural vibrations of the "spindle-workpiece-tool" subsystem of a lathe under static loading of the workpiece from a tool are carried out to verify the developed dynamic model of this machine tool, as well as to prove the validity of the model and its adequacy. The dynamic system characteristics are uniquely determined by the natural frequencies of its partial subsystems. This characteristic is chosen to prove the adequacy of the developed model. A necessary and sufficient condition for proving the adequacy of the developed dynamic model is the closeness of the experimentally measured and calculated natural frequencies of the model.

There is no doubt that many factors have a significant influence on the real machining process, and of course, the entire machine structure is involved in the vibrations that occur during machining. This is shown in many publications. However, the paper is devoted to the general principles of a model building of closed (through the cutting process) dynamic systems of the machining process using the method MSTMM (transfer matrix method of multibody systems). The developed model is part of the general model of the machine dynamic system. But it determines the main dynamic characteristics that form the dynamic quality of the entire machine mechanical structure, in particular, its vibration resistance. The studies of the influence of different physical factors on the vibration formation of the entire mechanical structure of the machine tool and the formation of the real machining quality were not considered in the presented for consideration paper.

The authors planned to expand the developed dynamic model to include all parts of the machine mechanical structure. Experimental testing of the extended model during real machining is also planned.

The authors have added to the paper (section 2) information about the basic principles used in the development of the dynamic model of a metal-cutting machine tool and experimental testing of the developed model. All changes carried out in the paper are marked in red font.

At the same time, the following points should be considered in the "spindle-workpiece" system:

  1. Regarding the issue of cutting width, it is recommended that the author provide a more clear explanation and description in the paper. The author can provide more detailed experimental data and calculation processes to demonstrate the rationality of recalculating the width as 1mm in the cutting model and numerical simulation stage to avoid errors that may affect the conclusions of the paper.
  • Determination of cutting forces by direct measurement, calculation by analytical cutting model, or numerical simulation using a finite-element cutting model is performed in the proposed paper during free orthogonal (quasi-orthogonal, since the workpiece is not a parallelepiped but a round rod) cutting. To ensure free orthogonal cutting, the tool width must be greater than the workpiece width. The stress-strain state of the machined material during orthogonal cutting is flat, and one of the three principal stresses is, by definition, equal to 0. In this case, according to the definition of the free orthogonal cutting process, the main stress directed along the workpiece width and tool width is zero (see, for example, Zorev [60], Oxley [61], etc.). Thus, the cutting forces during orthogonal cutting are directly proportional to the workpiece (tool) width. This makes it possible to move from one value of the workpiece width to another value of this width. This makes it possible to move from one value of the workpiece width to another value of this width. That is, it enables the cutting forces measured at one width of the workpiece to be used to determine the cutting forces for a different workpiece width. To do this, the value of the measured or any other defined cutting forces must be multiplied by the ratio of the workpiece widths. Often in general the value of cutting forces referred to the workpiece width is used, i.e., with a workpiece width of 1 mm. For example, such units of cutting forces ([N/mm]) are used in many analytical cutting models (analytical cutting models of Time, Merchant, Zorev [60], Kinzle, Oxley [61], etc.).

The same unit of cutting forces measurement is used in the proposed paper to make it possible to compare cutting forces determined by direct measurement according to the analytical cutting model and according to the FEM model. It is only possible to compare measurement and calculation results if the cutting widths are the same in each case. At the same time, the finite-element model of cutting is limited by the counting time, which is proportional to the number of elements. In the FEM model, the cutting width is assumed to be 1 mm in order to keep the counting time manageable. Therefore, when comparing the cutting forces determined by different methods, they are reduced to the same cutting width equal to 1 mm. During the experimental research to determine the natural frequencies, the tool was pressed to the workpiece with a force corresponding to the accepted cutting width (width of the shoulder) and equal to 3 mm, i.e., a thrust force equal to 750 N (250N x 3 mm).

  1. Regarding the influence of spindle thermal deformation and tool wear on frequency, it is suggested that the author conduct more in-depth research and analysis on these factors, or explain them in the paper to fully consider the degree of influence of these factors, ensuring the accuracy and reliability of the conclusions.
  • Thermal deformation of the spindle and tool wear undoubtedly have an effect on the machining process. However, this influence concerns the development of forced vibrations, not the change of natural vibrations. In addition, thermal deformation and wear are not fast processes similar to dynamic processes. Therefore, thermal deformations can affect the frequencies of natural vibrations of mechanical elements of a dynamic system only if the stiffness or mass of individual elements of this system is significantly changed due to these deformations. Changing the spindle unit stiffness is possible by heating the bearings of its supports, but it will not significantly affect the dynamic characteristics (natural frequencies). The same applies to the results of tool wear, accompanied by an insignificant change in its mass and stiffness.

  1. Regarding the issue of the influence of vibration frequency in the measurement and calculation of "spindle-workpiece" vibration, it is recommended that the author provide more data and analysis in the paper to further demonstrate the validity and accuracy of the conclusions. For example, the author can compare experimental data under different conditions to demonstrate the influence of vibration frequency on pressure and tool position and explore the influence of a single variable on the conclusions. In addition, it is suggested that the author clearly state the limitations and room for improvement of the conclusions to encourage further research and exploration.
  • The results of theoretical and experimental studies were compared by analyzing the natural vibration frequencies rather than the frequencies of forced vibrations. In planning the experimental studies, the authors took into account the most significant factors affecting changes in the dynamic characteristics (natural vibration frequencies) of the "spindle-workpiece-tool" system: changing mass-inertial characteristics and the conditions of contact interaction between the workpiece and tool. To account for the effect of mass, the experimental studies were carried out for two types of workpieces (solid and tubular), significantly different in mass, and - insignificantly different in stiffness. To evaluate the effect of contact interaction between the workpiece and tool, experimental studies were carried out in the absence of contact interaction, and with changes in the point coordinate of the load application (pressed with a cutter). To increase the effect of changing the point coordinate of the load application - the study was carried out on the cantilevered workpiece. As a result, the authors identified specific patterns of formation and changes in the values of natural vibration frequencies of the workpiece and spindle, reported in chapter 2 “Discussion”. Studies of experimental studies of forced vibrations arising in the cutting process are beyond the scope of the proposed paper and are the objective of further research.

In the paper text was added the information (see chapter 2) about the basic principles used in the development of the dynamic model of a metal-cutting machine tool and experimental testing of the developed model.

  1. Comments on the Quality of English Language

Some English sentences and grammar can be improved in the total paper.

  • English level checked and some corrections have been performed as recommended by the reviewer.

All changes carried out in the paper are marked in red font.

Reviewer 2 Report

Reviewer comments 

This paper stated on the “Cutting process consideration in dynamic models of machine tool spindle units”.

- The title of paper is not clear. machine tool spindle units of what?

- The quality of paper is good. The authors have described well about their work.

However, the length of paper is too long. The length of paper should be reduced. It is very difficult to find the main highlight from this paper.

- Minor editing of English language required.

Author Response

The authors are very grateful to Reviewer #2 for meticulously reviewing and interpreting the content of the paper.

Reviewer #2:    This paper stated on the “Cutting process consideration in dynamic models of machine tool spindle units”.

  1. The title of paper is not clear. machine tool spindle units of what?
  • The title of the paper specifies that the spindle assemblies of metal-cutting machines are considered. The paper proposes and studies the method of developing dynamic models of spindle assemblies of practically well-known metal-cutting machines. Experimental studies to confirm the proposed method were performed for the lathe spindle assembly.
  1. The quality of paper is good. The authors have described well about their work.
  • The authors are grateful to the reviewer for such an evaluation of the paper.
  1. However, the length of paper is too long. The length of paper should be reduced. It is very difficult to find the main highlight from this paper.
  • Probably the paper seems to be rather lengthy. However, this is due to the need for a complete, consistent, and logical presentation of the paper material. Section "2.2.1. Model of the spindle unit system" briefly describes the method used to create a dynamic model of the spindle units for machine tools. Section "3.1. Dynamic Model of System "Spindle Unit" outlines the theoretical basis of the development method of the dynamic model for spindle units. Section "3.4. Determination of natural frequencies vibration of the lathe elastic system "spindle-workpiece-tool" is devoted to the concrete application of the proposed method in developing a dynamic model of the particular lathe spindle. In the same section, the results of the measurement and calculation of natural vibration frequencies of the "spindle-workpiece-tool" subsystem and their comparison are presented. Section "4. Discussion" contains a discussion of the results of the performed experiments and calculations. The main results of the studies are presented in Sections 2.2.1, 3.1, and 3.4. Reducing the presented material of any section or sections will lead to a violation of its completeness and presentation logic. In this case, it will be almost impossible for non-particularly trained readers to understand the proposed method and the research results obtained. Those parts of the paper that could be separated from the main material have been placed in the appendices.
  1. Comments on the Quality of English Language

- Minor editing of English language required.

  • English level checked and some corrections have been performed as recommended by the reviewer.

All changes carried out in the paper are marked in red font.

Reviewer 3 Report

The article is written in a good scientific style, the analysis of the literature is presented extensively, exhaustively referring to the scientific scope of the work. The article requires several significant corrections and clarifications, in particular regarding the research experiment.
In the material for the experiment section, the lathe is indicated: "all-purpose engine lathe GH1230", the designation of the lathe as an engine lathe does not fit into the classification and division of lathe applications, the designation itself does not specify anything, please indicate the model and manufacturer of the machine and the year of production.
line 157, it is indicated that the shafts are made of 1045 steel, but how is the sleeve made? (tubular piece) of the same material?? if so, how, parent material of all samples must be the same.
Figure 1 is quite chaotic and raises doubts as to the practical experiment, the author accurately describes the geometry of the tool, giving the material designation - plate, the presented tool is a uniform, older construction.
Please present the experiment and its description with correctness.
feed rate parameters were not indicated - please complete it
overhang of the workpiece above 4 diameters (30/122mm) and machining without support will result in damage to the tool as the author achieved such a result, the machine used - visible in the photo will not provide additional stable working conditions
Figure 10 shows the hardness given in the MPa unit, such an assumption is incorrect,
The discussion chapter refers to the results quite exhaustively, unfortunately the conclusions are succinct, they do not refer to the results obtained, they are rather decorative. They are obvious from the point of general theses related to machining - in particular, line 773-776
Please list the key achievements in relation to the results as bullet points (5-6)

language in the technological area requires clarification and consistent use throughout the work

Author Response

The authors are very grateful to Reviewer #3 for meticulously reviewing and interpreting the content of the paper.

Reviewer #3:    The article is written in a good scientific style, the analysis of the literature is presented extensively, exhaustively referring to the scientific scope of the work. The article requires several significant corrections and clarifications, in particular regarding the research experiment.

  1. In the material for the experiment section, the lathe is indicated: "all-purpose engine lathe GH1230", the designation of the lathe as an engine lathe does not fit into the classification and division of lathe applications, the designation itself does not specify anything, please indicate the model and manufacturer of the machine and the year of production.
  • Information on the machine model, manufacturer, and year of manufacture has been added to the text of the paper as recommended by the reviewer. All changes carried out in the paper are marked in red font.
  1. line 157, it is indicated that the shafts are made of 1045 steel, but how is the sleeve made? (tubular piece) of the same material?? if so, how, parent material of all samples must be the same.
  • The workpieces from AISI 1045 steel were purchased as both rods and tubes (Manufacturer: Mannesmann Precision Tubes GmbH). The outside diameter of the material purchased as a tube was 32 mm, and the inside diameter was 24 ± 0.15 mm. After cutting the workpieces to the required length, they were pre-machined by longitudinal turning along the outside diameter to the size of 30 mm. This information has been added to the paper.
  1. Figure 1 is quite chaotic and raises doubts as to the practical experiment, the author accurately describes the geometry of the tool, giving the material designation - plate, the presented tool is a uniform, older construction.
  • The authors very much ask the respected reviewer to disclose the wording "chaotic" of Figure 1. In addition, the authors very much ask to inform what exactly are the reviewer's doubts about the practical experiment. In their description of the tool, the authors used the term "carbide plate" (line 202) rather than "insert". The term "carbide plate" refers to the common term for a cutting element, regardless of how it is mounted. The term "insert" implies a "clamped changeable cemented carbide insert”. The carbide plate mentioned in section "2.1 Materials" and used in the experimental studies was brazed onto the cutter body (see Figure 1). Tool geometry was ensured by grinding the cutter rake and clearance faces after the carbide cutting plate was brazed on.

The authors ask the respected reviewer to note that Figure 1 shows "Experimental setup for measuring free damped vibrations" (see the caption to Figure 1). When measuring free damped vibrations, there was no movement of machine parts with a workpiece clamped in the chuck. The tool was pressed to the workpiece with a force corresponding to the thrust force during free orthogonal cutting and positioned at a distance Li from the face of the chuck. A description of this experiment, shown in Figure 1, is given in the paper (see lines 221-238). During the pre-machining of workpieces along the outside diameter by longitudinal turning, as well as during the formation of the shoulder and measuring of cutting forces during free orthogonal cutting (see the description of the experiment in lines 201-218), the workpiece was pressed by the machine's tailstock due to its significant overhang. The process of longitudinal turning and the process of free orthogonal cutting are not shown due to their triviality.

4. Please present the experiment and its description with correctness: feed rate parameters were not indicated - please complete it; overhang of the workpiece above 4 diameters (30/122mm) and machining without support will result in damage to the tool as the author achieved such a result, the machine used - visible in the photo will not provide additional stable working conditions

  • In free orthogonal cutting, the depth of cut plays a role of the cutting feed (see, for example, Zorev [60], Oxley [61], Tsekhanov [49], etc.). The depth of cut is given in the paper (see line 208): ap = 0.2 mm. Figure 1 shows the "Experimental setup for measuring free damped vibrations" (see figure caption). When measuring free damped vibrations, the movements of machine parts with the workpiece clamped in the chuck were of course absent. In this regard, there was also no need for the workpiece to be clamped at the tailstock side of the machine. However, during the workpiece machining along the outside diameter by longitudinal turning, as well as the formation of the shoulders and measuring of cutting forces during free orthogonal cutting, the workpiece was of course clamped by the machine's tailstock due to a significant overhang of the workpiece.
  1. Figure 10 shows the hardness given in the MPa unit, such an assumption is incorrect,
  • By definition, hardness is calculated as the ratio of the force applied to the indenter to the indentation area when the indenter penetrates the test material. In Brinell hardness testing, hardness is determined by the diameter of the impact left by a carbide ball pressed into the surface of the test material. The hardness value is calculated as the ratio of the force applied to the ball to the area of the impact (the area of the impact is taken as the area of the sphere part). Thus, the unit of hardness is kgf/mm2 (see e.g. https://material-properties.org/units-of-hardness-hardness-numbers-definition/). And the unit of hardness for static methods of its measurement is the same for the well-known methods of hardness determination of metals: the Brinell, Rockwell, Vickers, etc. (see ibid.). However, the unit of hardness is very often, and especially in industry, omitted and only the method of hardness measurement (HB, HR, HV or HBC, HRC, HVC) is indicated, e.g. 180 HB. In reality, it means that the hardness is 180 kgf/mm2 and is measured by the Brinell method. However, the international SI system of units uses N/m2 instead of kgf/mm2, which is Pa (Pascal). If we convert the hardness of 180 kgf/mm2 from non-system hardness units to the SI system, i.e. N/m2 or Pa, we get 1800 MPa (HB=1800 MPa). This is what the authors have shown in Figure 10.

6. The discussion chapter refers to the results quite exhaustively, unfortunately the conclusions are succinct, they do not refer to the results obtained, they are rather decorative. They are obvious from the point of general theses related to machining - in particular, line 773-776. Please list the key achievements in relation to the results as bullet points (5-6).

  • The major achievements obtained as a result of the studies presented in the paper are formulated in the form of key points, as recommended by the reviewer. To formulate the major achievements, the research results outlined in both the chapter “Discussion” and the chapter “Results” are used.

7. Comments on the Quality of English Language

language in the technological area requires clarification and consistent use throughout the work.

  • English level checked and some corrections have been performed as recommended by the reviewer.

All changes carried out in the paper are marked in red font.

Round 2

Reviewer 3 Report

Regarding the responses submitted:

(3) Line 205 - "The width of the carbide plate was 5 mm to ensure free orthogonal cutting" Line 199 - "The shoulder and groove width were 3 mm (see Figure 1)." Please explain how a 5mm wide cutting insert can make a 3mm groove?

(4) the author's explanations are purely theoretical. The basic technological parameter is the feed rate, which is synchronous in turning, and must be declared on the machine tool in the case of mechanical feed,...... what feed rate was determined for the process, it is important because it is directly related to the cutting forces and the impact on generating vibration. Please specify whether the machine tool used in the process has a mechanical feed. I do not want to lecture the authors on the kinematics of work on a lathe. I assume that my question is understandable

(5) Teaching and lecturing on hardness determination is not very professional, referring to a website and instructing a reviewer on this is rude. The author's lecture is very accurate from a physical point of view. However, in the technical area, I recommend the author to familiarize himself with the ISO-6506-1 standard and make a consistent markings in the article. The unit of hardness is omitted because in industrial realities we use standards.

Author Response

Answers to the comments of Reviewer #1:

The authors are very grateful to Reviewer #1 for meticulously reviewing and interpreting the content of the paper.

Reviewer #1:

  1. (3) Line 205 - "The width of the carbide plate was 5 mm to ensure free orthogonal cutting" Line 199 - "The shoulder and groove width were 3 mm (see Figure 1)." Please explain how a 5mm wide cutting insert can make a 3mm groove?
  • The grooves formation for measuring the cutting forces during free orthogonal cutting and measurements of free damped vibrations were carried out at the stage of main experiments preparation. The grooves were turned with an ordinary cutoff cutter. The insert width of the cutoff cutter was 3 mm. The relevant information is included in the paper (see lines 200-201).
  1. (4) the author's explanations are purely theoretical. The basic technological parameter is the feed rate, which is synchronous in turning, and must be declared on the machine tool in the case of mechanical feed,...... what feed rate was determined for the process, it is important because it is directly related to the cutting forces and the impact on generating vibration. Please specify whether the machine tool used in the process has a mechanical feed. I do not want to lecture the authors on the kinematics of work on a lathe. I assume that my question is understandable
  • During free orthogonal cutting process, the feed was f=0.2 mm/rev. The paper's text is corrected in the place of cutting parameters description (see line 207). The Gear Head Lathe GH1230 machine used for the experimental studies has a mechanical longitudinal and transverse feed.
  1. (5) Teaching and lecturing on hardness determination is not very professional, referring to a website and instructing a reviewer on this is rude. The author's lecture is very accurate from a physical point of view. However, in the technical area, I recommend the author to familiarize himself with the ISO-6506-1 standard and make a consistent markings in the article. The unit of hardness is omitted because in industrial realities we use standards.
  • The hardness of the machined material is corrected in Figure 10 and in the text of the paper (see line 198) as recommended by the reviewer.

In their response, the authors have attempted to explain why they used physically correct units of measurement instead of standardized Brinell hardness units. This was the sole purpose of the response. The authors regret that they chose a form of response that the esteemed reviewer interpreted as teaching and lecturing.

Round 3

Reviewer 3 Report

  The article may be forwarded for further processing